# High-resolution in situ structures of mammalian respiratory supercomplexes

Wan Zheng[1,2], Pengxin Chai[2], Jiapeng Zhu[1✉] & Kai Zhang[2✉]

Mitochondria play a pivotal part in ATP energy production through oxidative phosphorylation, which occurs within the inner membrane through a series of respiratory complexes[1–4]. Despite extensive in vitro structural studies, determining the atomic details of their molecular mechanisms in physiological states remains a major challenge, primarily because of loss of the native environment during purification. Here we directly image porcine mitochondria using an in situ cryo-electron microscopy approach. This enables us to determine the structures of various high-order assemblies of respiratory supercomplexes in their native states. We identify four main supercomplex organizations: $I_1III_2IV_1$, $I_1III_2IV_2$, $I_2III_2IV_2$ and $I_2III_4IV_2$, which potentially expand into higher-order arrays on the inner membranes. These diverse supercomplexes are largely formed by 'protein–lipids–protein' interactions, which in turn have a substantial impact on the local geometry of the surrounding membranes. Our in situ structures also capture numerous reactive intermediates within these respiratory supercomplexes, shedding light on the dynamic processes of the ubiquinone/ubiquinol exchange mechanism in complex I and the Q-cycle in complex III. Structural comparison of supercomplexes from mitochondria treated under different conditions indicates a possible correlation between conformational states of complexes I and III, probably in response to environmental changes. By preserving the native membrane environment, our approach enables structural studies of mitochondrial respiratory supercomplexes in reaction at high resolution across multiple scales, from atomic-level details to the broader subcellular context.

Mitochondria are central to energy production in eukaryotic cells[1]. Mitochondrial dysfunctions are associated with a broad range of severe ailments, including metabolic, cardiovascular, neurodegenerative and neuromuscular diseases[2–4]. Eukaryotic mitochondria comprise more than a thousand proteins, which dynamically assemble into complexes of various forms[5–7]. A key group of these are the respiratory chain supercomplexes (SCs); these which primarily consist of complexes I, $III_2$ and IV (CI, $CIII_2$ and CIV) in varying stoichiometries, which serve as the minimal functional units for $NADH:O_2$ oxidoreduction[8]. Recent studies in cryo-electron microscopy (cryo-EM) have determined structures of several mammalian respiratory complexes[9–16] and SCs[17–20], derived from in vitro purified proteins, revealing their subunit compositions, conformational dynamics and ligand-binding sites. Mechanistically, these structures have shed light on electron transfer pathways, proton-pumping mechanisms and regulatory elements[9–20]. Cryo-EM has also provided insights into possible assemblies within mitochondria and hypothetical functional roles of these respiratory SCs[17–21]. However, traditional in vitro approaches result in a loss of the native environment, which poses a significant challenge in elucidating the actual assemblies and molecular mechanisms under physiological conditions.

Here we report in situ structures of porcine respiratory SCs, determined directly through imaging porcine mitochondria by single-particle analysis combined with cryo-electron tomography (cryo-ET) (Fig. 1, Extended Data Figs. 1 and 2, Supplementary Figs. 1–5, Supplementary Video 1 and Supplementary Tables 1–6). Our structures, with an average resolution of approximately 2.5 Å and local resolution up to 1.8 Å in the best regions, enable in-depth study of mitochondrial respiratory SCs in their native states. With this resolution, we resolve numerous reactive intermediates within these SCs and determine structures of the four main types of SC organization.

Our structures show distinct classes with different density patterns for endogenous ubiquinone/ubiquinol ($Q/QH_2$) and its interacting residues, providing new structural insights into the $Q_{10}$ exchange dynamics within CI. We also capture multiple states of $Q_{10}$ bound to the $Q_o$ sites of $CIII_2$, along with positional shifts in the Rieske domain. Our high-resolution maps unambiguously show complex hydrogen-bond networks, offering detailed perspectives on proton transfer and the Q-cycle mechanism.

To assess the impact of pathogenic conditions on these complexes, we subjected porcine hearts to various treatments mimicking different levels of myocardial ischaemia, which affect the distribution of reactive

[1]School of Medicine & Holistic Integrative Medicine, Nanjing University of Chinese Medicine, Nanjing, China. [2]Department of Molecular Biophysics and Biochemistry, Yale University, New Haven, CT, USA. ✉e-mail: zhujiapeng@hotmail.com; jack.zhang@yale.edu

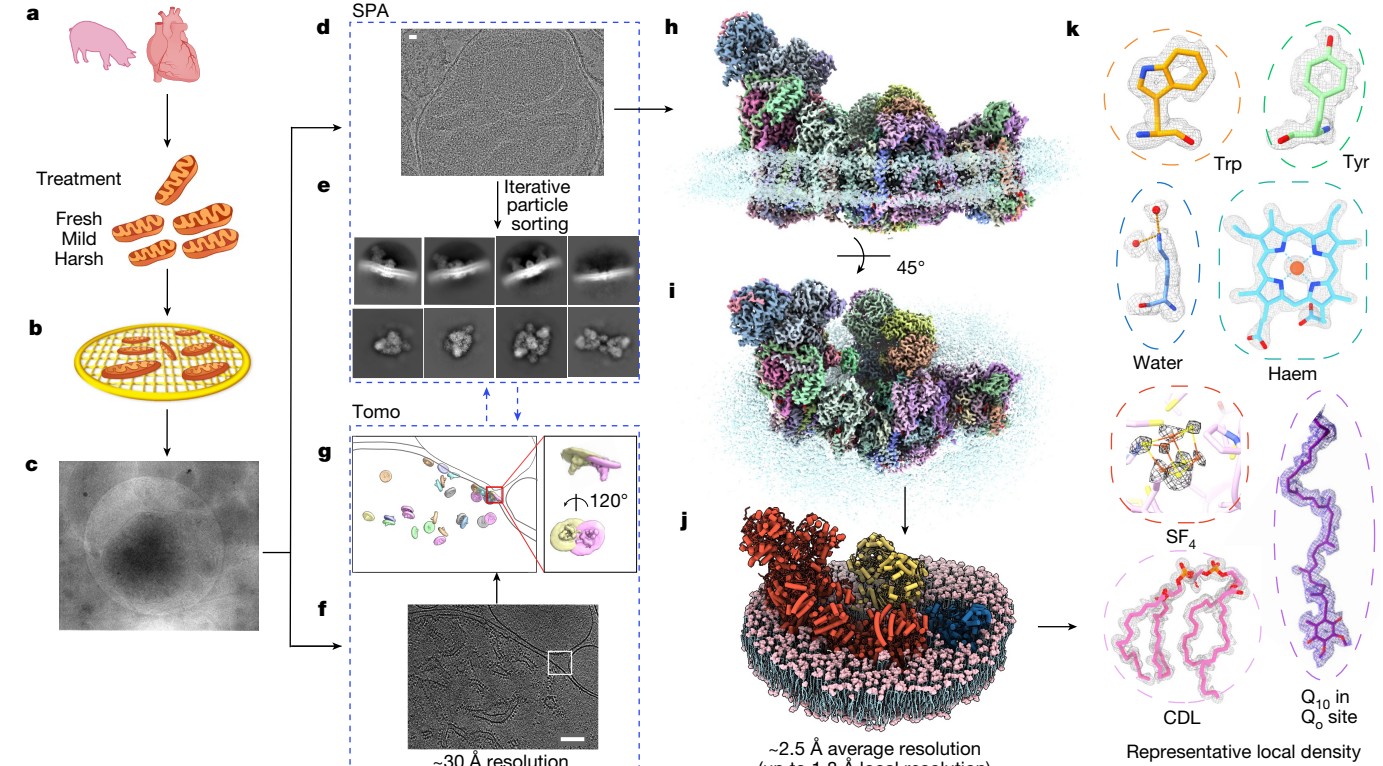

**Fig. 1 | In situ single-particle cryo-EM and cryo-ET analysis of mammalian mitochondrial respiratory SCs. a,b**, Grid preparation of porcine mitochondria. Mitochondria were extracted from porcine hearts treated under different conditions (fresh, mild and harsh) (**a**) and directly frozen on to cryo-EM grids (**b**) for subsequent in situ imaging. **c**, A representative image of porcine mitochondria under low magnification. Hole diameter, 2 μm. **d**, Representative cryo-EM micrograph of mitochondria for single-particle analysis (SPA). Scale bar, 20 nm. **e**, Representative 2D class averages showing different types of SC after 3D classification. **f,g**, Representative tomographic slices (**f**) and corresponding subvolume averages (**g**) of a reconstructed tomogram (Tomo). Scale bar, 100 nm. **h,i**, Side (**h**) and top (**i**) views of a representative high-resolution map of an SC in the native mitochondrial inner membrane. **j**, A molecular model of the high-resolution SC with the surrounding membrane built. **k**, High-resolution features shown by representative density of amino acid residues and endogenous ligands. CDL, cardiolipin.

states in respiratory complexes without compromising the resolution of the cryo-EM maps. Overall, our in situ approach enables investigation of the impacts of diverse mitochondrial diseases and pharmacological treatments by determining reactive protein structures under physiological conditions within mitochondria.

## Four main SC types in situ

Three-dimensional (3D) reconstruction unveiled diverse forms of SCs with distinct compositions, including four dominant types: $I_1III_2IV_1$ (type A), $I_1III_2IV_2$ (type B), $I_2III_2IV_2$ (type O) and $I_2III_4IV_2$ (type X) (Fig. 2a, Extended Data Fig. 2 and Supplementary Figs. 2–5). Types A and O are similar to previously reported in vitro structures[17–20], whereas types B and X represent two new forms not observed in vitro. Our cross-classification results indicate that the type-A SC (Fig. 1h–j) is the most abundant form, determined at resolutions of 1.8–2.4 Å in most CI and $CIII_2$ regions (Supplementary Figs. 2 and 6) and at approximately 2.75 Å in CIV (Supplementary Fig. 2). The high-quality maps, as demonstrated by discernible 'holes' in the side chains of aromatic residues, the precise positions of pyrrole rings and the Fe atom in haem, and S/Fe atoms in $SF_4$, enabled us to build accurate atomic models of this form (Fig. 1k, Supplementary Fig. 6 and Supplementary Video 2). For the type-A SC, we modelled 6,327 water molecules into the structure (Extended Data Fig. 3), including those that were likely to have central roles in proton transfer across the membrane. Furthermore, we built 197 structured and associated lipids (Fig. 2c, Extended Data Fig. 4 and Supplementary Video 3) that contribute to stabilization of the protein structure, enhancing the stability of the SCs by facilitating essential protein–lipid–protein interactions (Fig. 2c,d), creating a hydrophobic environment in the Q-binding sites (Extended Data Fig. 4b and Supplementary Video 3) and participating in the hydrogen-bond network through their polar heads (Extended Data Fig. 5 and Supplementary Video 4). We directly visualized mitochondrial inner membranes composed of more dynamic lipids surrounding the proteins and built atomic models (Fig. 2c, Extended Data Fig. 4 and Supplementary Video 3). Although the overall architecture resembles a previously reported structure using in vitro purified protein[17], our high-resolution in situ structure shows substantial differences at the interaction interfaces among complexes I, $III_2$ and IV (Supplementary Fig. 7). Previous work has demonstrated that the lipid bilayer can bend to adapt to CI conformations[22]. Consistent with these findings, the membrane enveloping the SC exhibited noticeable curvature, varying across different regions (Fig. 2b). Specifically, the membrane at the CI heel bends towards the mitochondrial matrix, whereas the $CIII_2$ region displays the opposite curvature pattern (Fig. 2b, Extended Data Fig. 6 and Supplementary Video 5).

Our high-resolution density maps of native respiratory SCs allow for accurate analysis of the interaction interfaces. In type-A SC (Fig. 2a), CI–$CIII_2$ interactions are mediated through specific contacts between the NDUFA11[I] and UQCRQ[III] subunits, as well as NDUFB9[I], NDUFB4[I] and UQCRC1[III]. For CI–CIV interactions, the key participating subunits are NDUFB3[I], ND5[I] and COX7A[IV] (Supplementary Fig. 7). One striking feature of the in situ structure is that the interstitial space among CI, $CIII_2$ and CIV is populated by lipids that mediate the complex interactions.

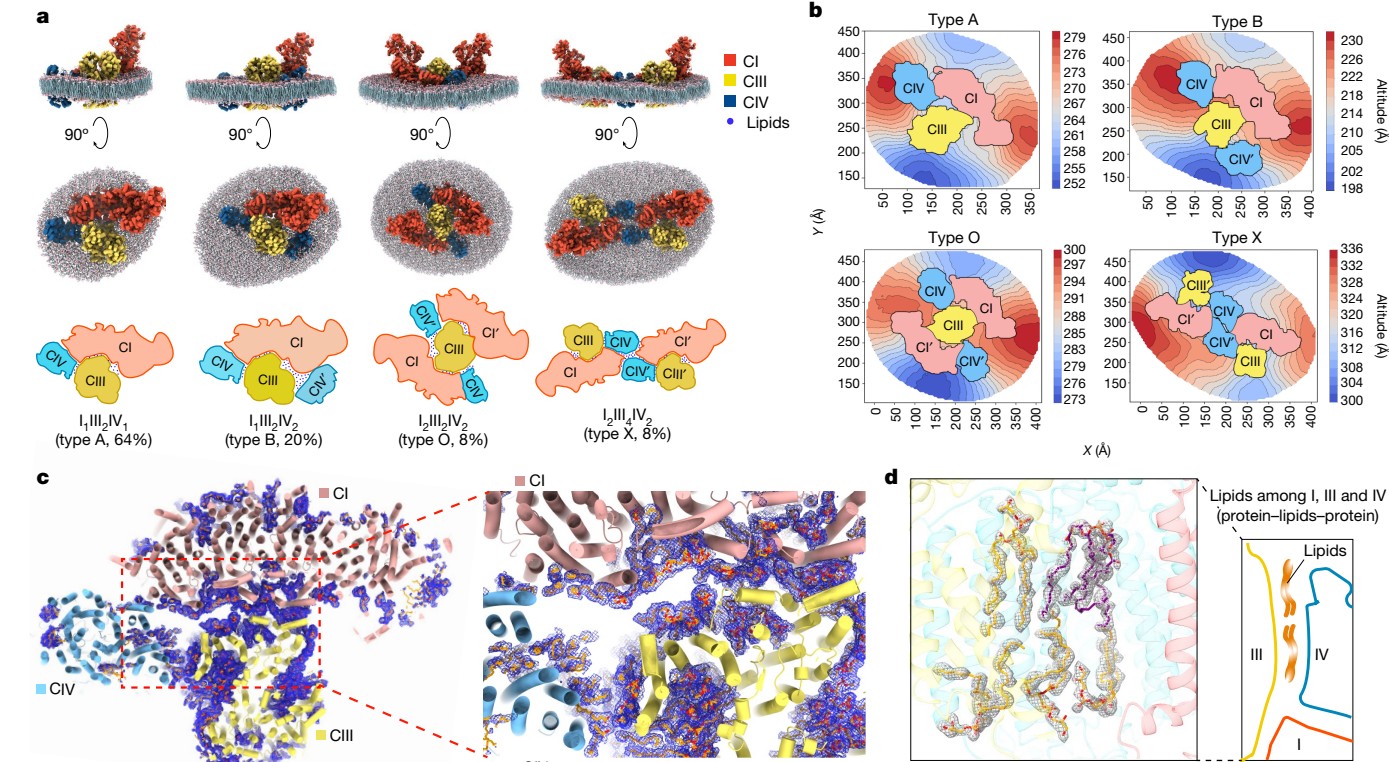

**Fig. 2 | Architecture, membrane curvature and interaction interfaces of the four types of respiratory SC. a**, Top and side views and cartoon models of SCs $I_1III_2IV_1$, $I_1III_2IV_2$, $I_2III_2IV_2$ and $I_2III_4IV_2$ with models of the surrounding membranes. These views highlight the impact of different SC compositions on local membrane curvature. **b**, Contour maps of the native membrane around the four main types of SC, viewed from the mitochondrial matrix side. Red and blue indicate high and low altitudes, respectively. These gradients clearly demonstrate the common feature in which the membranes surrounding the CI heel and $CIII_2$ regions are convex and concave, respectively. They also show distinct local curvature differences among the four types. **c**, Interaction interfaces among CI (pink), $CIII_2$ (light yellow) and CIV (light blue) in type-A SC. Blue mesh represents the density maps of lipid molecules filling the interstitial space among these complexes. **d**, Representative local density maps and atomic modes of lipids built in the interface between $CIII_2$ and CIV.

In particular, no direct protein–protein interactions between $CIII_2$ and CIV were observed in our structures (Fig. 2d).

In contrast to type A, the type-B SC (Fig. 2a) incorporates a second CIV (CIV′) situated between $CIII_2$ and the hydrophilic arms of CI. Unlike $CIII_2$, which exhibits two-fold symmetry, this extra CIV in the type-B SC is not two-fold symmetric to the first CIV relative to the symmetry axis of $CIII_2$. Instead, it displays an approximately 60° rotation (Supplementary Fig. 8a,b), partially enclosing the lipid bilayer between the Q-binding pockets in CI and $CIII_2$ (Supplementary Fig. 9 and Supplementary Video 6) and forming an architecture that could confine free diffusion of $Q_{10}$ and facilitate Q-channelling. In type-B SC, CIV′ establishes new interactions with CI through the NDUFA1[I] and COX5A[IV′] subunits and with $CIII_2$ through the UQCR10[III] and COX6A[IV′] subunits (Supplementary Fig. 7).

In addition, two higher-order SCs were determined through a multilevel cross-classification approach we developed. The $I_2III_2IV_2$ (Fig. 2a) complex at 2.6–3.3 Å resolution (Supplementary Fig. 4), designated type O owing to its overall shape, resembles the structure of the human mitochondrial megacomplex[20] (Supplementary Fig. 8e) and can be considered to be a pseudo-$C_2$ symmetric expansion of type A, sharing the $CIII_2$ dimer (Supplementary Fig. 8c). In contrast to the detergent-purified type-O SC, we observed that the native type-O SC exhibits pseudo-$C_2$ rather than strict $C_2$ symmetry, with the surrounding membrane bending towards the mitochondrial matrix. This observation is not surprising given the highly curved membrane structures of the cristae.

The new SC $I_2III_4IV_2$ (Fig. 2a), termed type X owing to its chromosome-like appearance, is formed by two type-A SCs aligned in a head-to-head manner along a pseudo-two-fold axis (Supplementary Fig. 8d).

No canonical strong protein–protein interactions were evident at the dimerization interfaces between the two type-A SCs, except for potential weak contacts between helices NDUFB3[I] and COX4[IV]. The dimerization is primarily mediated by 'protein–lipids–protein' interactions in our structure, as evidenced by the numerous lipid molecules filling the interfacial spaces (Supplementary Video 3).

Notably, in all the four main SC forms, CIV shows the most freedom in terms of its binding positions and modes of interaction with other complexes. Different isoforms of CIV subunits have been reported and may affect SC formation. However, using the porcine heart mitochondria, we only identified one specific set of CIV subunit isoforms (Supplementary Table 7), probably because their expression levels are highest in this tissue[23]. Moreover, our structural analyses consistently identified the NDUFA4 subunit as an integral part of CIV across the four types of SC, distinctly separated from CI (Supplementary Fig. 10); this clarifies the positional association of NDUFA4 and is consistent with the findings of a recent study[24].

Furthermore, the formation of these SCs substantially influences the local curvature of the surrounding membranes (Fig. 2b, Extended Data Fig. 6 and Supplementary Video 5). Vice versa, we speculate that the membrane geometry in turn affects the overall arrangement, distribution and conformation of these SCs; this is worth further investigation. In addition, we detected potential extensions of respiratory SCs into high-order arrays in the mitochondrial inner membranes.

## Multiple Q-binding states in CI

CI orchestrates electron transfer from NADH to ubiquinone and concurrently translocates protons across the inner mitochondrial

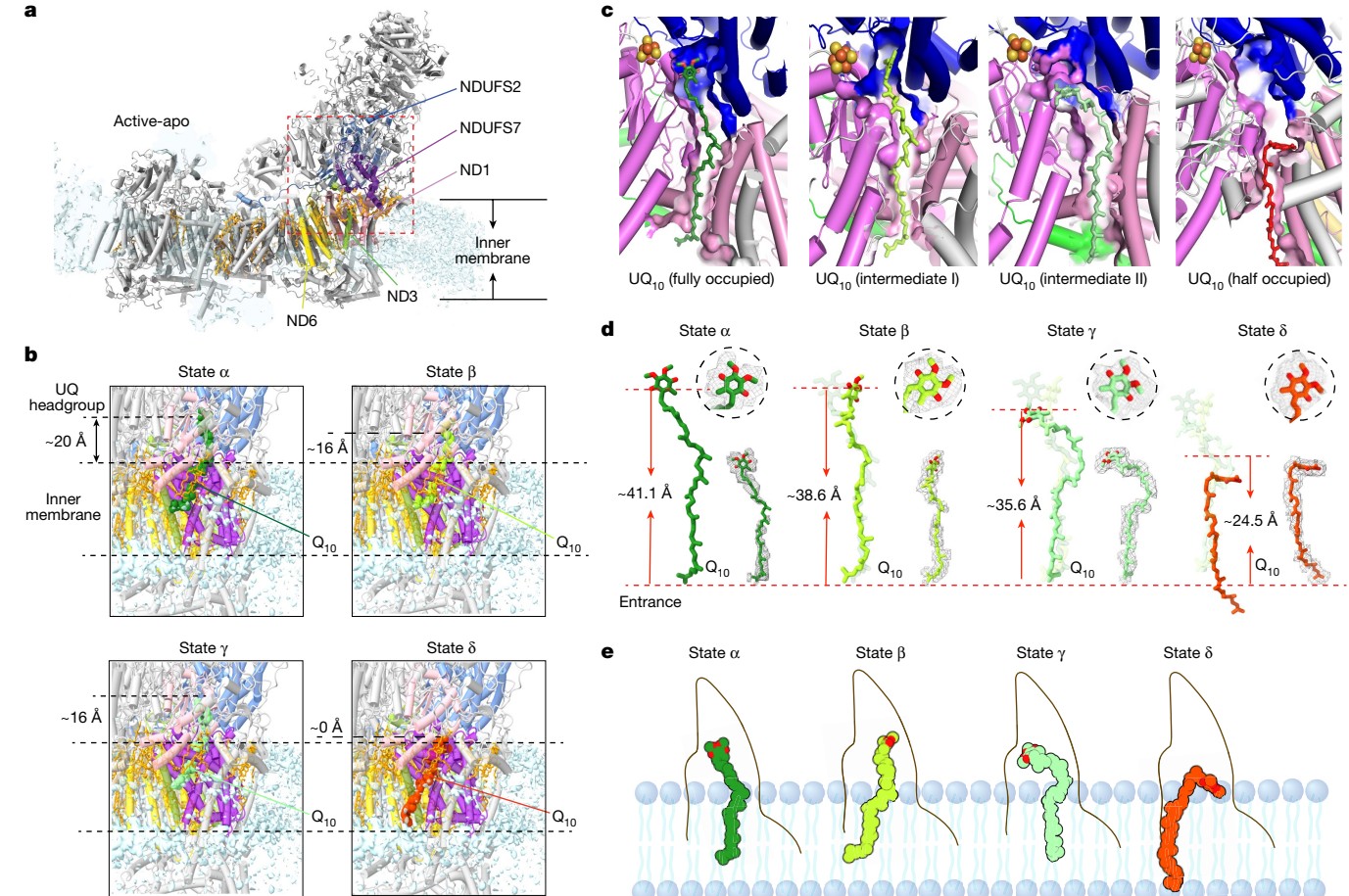

**Fig. 3 | High-resolution in situ structures reveal multiple Q/QH₂ binding states within the Q-binding channel. a**, Cartoon models of CI active-apo state, with subunits constituting the Q-binding channel highlighted. **b**, Variations in distances between the $Q_{10}$ headgroup and membrane surfaces across fully occupied, two intermediate, and half-occupied states. The height of the membrane surface was estimated using the average of lipid headgroups surrounding the CI 'heel'. **c**, Spatial positioning of $Q_{10}$ within the Q-binding channel for the four distinct binding states. **d**, Comparisons of three other binding states with the fully occupied $Q_{10}$ (transparent stick). The distance from the $Q_{10}$ headgroup to the quinone-binding channel entrance varied among different binding states. **e**, Schematic depiction of the silkworm-like undulatory motion of $Q_{10}$ within the Q-binding channel.

membrane[25]. The Q-binding site in CI features an unusually long, heterogeneous channel structure[26–28]. The entry, exit and interactions of $Q_{10}$ within this channel are central yet not fully resolved questions in our understanding of the molecular mechanisms of CI. Recent structural studies and molecular dynamic simulations have resolved distinct intermediates of Q-binding states[22,29].

We identified five main Q-binding states (Fig. 3), including the apo state and a previously described Q-bound state[28,29] (state α), along with three new Q-bound states (states β–δ) (Fig. 3b and Extended Data Fig. 7), which are different from the three Q-binding sites previously reported[29] (Extended Data Fig. 8). The apo state displays only noisy densities in the Q channel. State α resembles the active form of bovine and porcine CI (PDB: 7QSK[28], 7V2C[29]) (Extended Data Fig. 7b), with a $Q_{10}$ molecule fully occupying the Q-binding site (Fig. 3c). In this state, the 1-carbonyl group of $Q_{10}$ is situated approximately 20 Å (Fig. 3b) above the membrane surface (M-distance) and 41.1 Å (Fig. 3d) from the channel entrance (E-distance). The three other states show not only conformational differences in the $Q_{10}$ headgroup and the key residue H59^NDUFS2 (Extended Data Fig. 7a) but also long-range structural changes away from the $Q_{10}$ headgroup and along the whole $Q_{10}$. In state β, the Q headgroup is angled away from the Fe–S cluster $N_2$ relative to state α, although the tail region largely remains similar to that in state α (Fig. 3d). These conformational changes result in an M-distance of 16 Å (Fig. 3b) and an E-distance of 38.6 Å (Fig. 3d).

State γ is distinguished by housing a $Q_{10}$ molecule with a headgroup aligned more parallel with the membrane and a contorted midsection of the Q tail. Although the M-distance remains consistent with that in state β, the E-distance is reduced by 3 Å (Fig. 3d), a change attributable to its silkworm-like undulatory motion (Fig. 3e). In state δ, a $Q_{10}$ molecule only partially fills the Q channel (E-distance = 24.5 Å). Its headgroup aligns flush with the membrane surface (M-distance = 0), while the tail spans the entire lipid bilayer (Fig. 3b). Furthermore, the headgroup in state δ is surrounded by a cluster of acidic, basic and polar residues inside the Q channel (Extended Data Fig. 7). These residues seem to create a highly polarized environment connected to the aqueous matrix, facilitating Q protonation. Taken together, our findings support a model in which $Q_{10}$ progresses through the Q channel by peristaltic motion (Fig. 3e).

## Active–deactive transitions in CI

Under physiological conditions and without substrates, mammalian CI transitions from an active, ready-to-catalyse state to a substantially deactive resting state[28,30,31]. During ischaemia, a condition characterized by limited oxygen supply, the deactive state emerges owing to cessation of the electron transport chain[28]. To investigate the effects of varying levels of ischaemia on the atomic-level structure of CI in native mitochondria, we exposed porcine hearts to room temperature for

durations of 0 min ('fresh'), 40 min ('mild') and more than 4 h ('harsh') before mitochondrial isolation (Fig. 1b).

The active and deactive states of CI are delineated by distinct structural hallmarks, including domain movements and conformational alterations around the Q-binding site and in proximal membrane-domain subunits[28]. Using focused 3D classification, we analysed these hallmarks in the SC structures determined from mitochondria under the three conditions (Supplementary Figs. 11 and 12). In the fresh sample, approximately 75% of CI in SCs adopted an active state, compared with around 30% and 18% under mild and harsh conditions, respectively (Extended Data Fig. 9).

Previous in vitro structural studies have suggested that the hallmarks of CI can be collectively categorized into either the active or deactive form[26]. To further investigate whether there existed intermediate states involved in the transition from active to deactive (A–D transition), we performed further focused 3D classification targeting the regions around the Q site using all datasets combined, resulting in seven main intermediate classes. We built the models in these states and compared them with the canonical active and deactive structures (Supplementary Fig. 12 and Supplementary Table 8). The comparison unveiled two distinct classes (classes 0 and 7 in Supplementary Table 8) that fully conformed to the hallmarks of the active and deactive states, respectively. By contrast, other classes exhibited only a subset of the hallmarks associated with the deactive state, indicating that there may exist various stages in the transition between the two extreme states in the native mitochondrial environment.

## Catalytic states of CIII$_2$ in reaction

CIII$_2$ transfers electrons from ubiquinol to cytochrome $c$ (cyt $c$) and contributes to the proton gradient for ATP synthesis[32,33]. Comprising three core subunits (cyt $b$, cyt $c_1$ and the Rieske iron–sulfur protein (ISP)) along with eight auxiliary subunits, each CIII hosts four metal centres (haems $b_H$ and $b_L$ in cyt $b$, the [2Fe–2S] cluster in ISP and haem $c_1$ in cyt $c_1$) and two Q sites: Q$_i$ for Q reduction and Q$_o$ for QH$_2$ oxidation[34–36]. From our high-resolution maps, with local resolutions between 1.8 and 2.4 Å (Supplementary Figs. 2 and 6) except for the dynamic ISP, we unambiguously identified all reactive centres and built atomic models of all endogenous ligands (Fig. 4a,b and Extended Data Fig. 4). Focused 3D classification showed that endogenous Q$_{10}$ ligands bound to both Q sites adopt multiple conformations representing different reaction stages. Further well-defined densities in the Q-binding pockets were identified as structured lipids with their phosphate headgroups tightly bound to the surrounding protein regions. Tails of these lipids create a hydrophobic, dynamic environment for Q-binding and release.

During the Q-cycle, Q$_{10}$ at the Q$_i$ site undergoes a two-step reduction, acquiring two protons from the mitochondrial matrix to form QH$_2$ (ref. 37). Our structure showed a hydrogen-bond network near the Q$_i$ site, comprising water molecules (Fig. 4c,d, Extended Data Fig. 5 and Supplementary Video 4), polar amino acid residues and the phosphate heads of three lipids (two cardiolipin and one phosphatidylethanolamine) (Fig. 4d and Extended Data Fig. 5), which form a Grotthuss-competent system for proton transfer[25,38]. The network could be divided into two hydrogen-bond chains primarily constituted of water molecules that link Q$_{10}$ to the matrix (Supplementary Video 4). One of the two chains further branches above the three lipids that is directly involved in facilitating the proton transfer through the hydrogen network. This bifurcated chain engages two charged amino acids, K227$^{MTCYB}$ and D228$^{MTCYB}$, with D228$^{MTCYB}$ linked to the 1-carbonyl of Q$_{10}$ (Fig. 4c and Extended Data Fig. 5). The second chain, comprising 5 waters that surround H201$^{MTCYB}$ (Fig. 4c), is directly connected to 4-carbonyl of Q$_{10}$. These chains provide an explanation for how the rapid proton transfer process is achieved in the Q$_i$ site.

At the Q$_o$ site, QH$_2$ releases two protons into the intermembrane space as part of its reaction cycle[39]. Our high-resolution map clearly resolved a prevalent Q-binding conformation at this site, characterized by π–π stacking interactions with F274$^{MTCYB}$ (Fig. 4e). Similar to the Q$_i$ site, there was also a hydrogen-bonded water chain near the Q$_o$ site for proton transfer. Notably, residue E271$^{MTCYB}$, engaged in this water chain, exhibited clear dual conformations (Conf. 1 and Conf. 2) and is likely to play a critical part in proton transfer (Fig. 4e). The two conformations resembled those of two previously reported crystal structures in the apo state[40] (Conf. 1) and with the Q$_o$ inhibitor stigmatellin bound[41] (Conf. 2), respectively (Fig. 4f). Further structural comparison indicated that the headgroup of stigmatellin lies closer to the ISP than the endogenous QH$_2$ captured in our in situ structure. Focused 3D classification uncovered a less-populated (approximately 13%) Q-binding state (Fig. 4g) at the Q$_o$ site with its headgroup shifted around 6 Å closer towards the ISP compared with the dominant form (Fig. 4h,i). We propose that this state, rather than the dominant class, represents a transient electron-hopping phase from QH$_2$ to [2Fe–2S] in ISP.

The transfer of electrons from QH$_2$ at the Q$_o$ site to cyt $c_1$ is mediated by movement of the Rieske domain[33,37]. Although early structural studies have hinted at its dynamic role in electron transfer, questions remain about how this motion is initiated, regulated and coupled with Q-binding states during the catalytic cycle. In addition to the apo form, we resolved four distinct Q-bound states (Fig. 4h,i). In state I, the Q$_{10}$ headgroup is situated near the [2Fe–2S] cluster at a distance of 10.8 Å, with the Rieske domain exclusively adopting the b position. In states II, III and IV, Q$_{10}$ takes on the prevalent conformation revealed in our cryo-EM maps, while the Rieske domain undergoes a stepwise movement from the b position towards the c position, eventually making contact with cyt $c_1$ for electron transfer in state IV. To further analyse the detailed conformational changes among these states, we measured two characteristic distances, one from [2Fe–2S] to Q$_{10}$ and the other to haem $c_1$ (Fig. 4h). Our results indicate a possible sequential progression from state I to state IV, characterized by increasing [2Fe–2S]–Q$_{10}$ distances and decreasing [2Fe–2S]–haem $c_1$ distances. On the basis of these observations, we propose a model that outlines the coupling between ISP movement and Q-binding states.

## CI and CIII$_2$ conformational correlation

To examine the potential correlations between the conformational states of CI and CIII$_2$ in the same mitochondrial SC, we conducted focused conformational analyses of paired CI and CIII$_2$ from samples treated differently (Fig. 1b). In the fresh sample, CI predominantly existed in its active state; concurrently, the ISP domain of CIII$_2$ adopted the b position (Extended Data Fig. 9b). Under harsh conditions, most CI particles were classified into the deactive form, whereas the ISP domain of CIII$_2$ preferentially adopted the c position (Extended Data Fig. 9c). Notably, for the mild treatment, despite CI predominantly being in the deactive state, the ISP domain of CIII$_2$ primarily remained in the b position, indicating that it may maintains catalytic functionality for a short duration after deactivation of CI (Extended Data Fig. 9c). Previous research indicates that the deactive state of CI arises as a result of disrupted electron transfer through the respiratory chain owing to limited oxygen supply under ischaemic conditions[28]. High-resolution structural analyses using X-ray crystallography and single-particle cryo-EM also demonstrate that CIII$_2$ adopts the c position in the absence of substrate[40]. This behaviour of CI/CIII$_2$ conformational distributions could be attributed to direct effects of the treatments or to environmental changes induced by different treatments, potentially affecting the turnover of respiratory chain substrates.

## Discussion
### Possible functional roles of SCs

The existence of mammalian respiratory SCs has been established by both in situ cryo-ET[42] and in vitro approaches[17,18], including

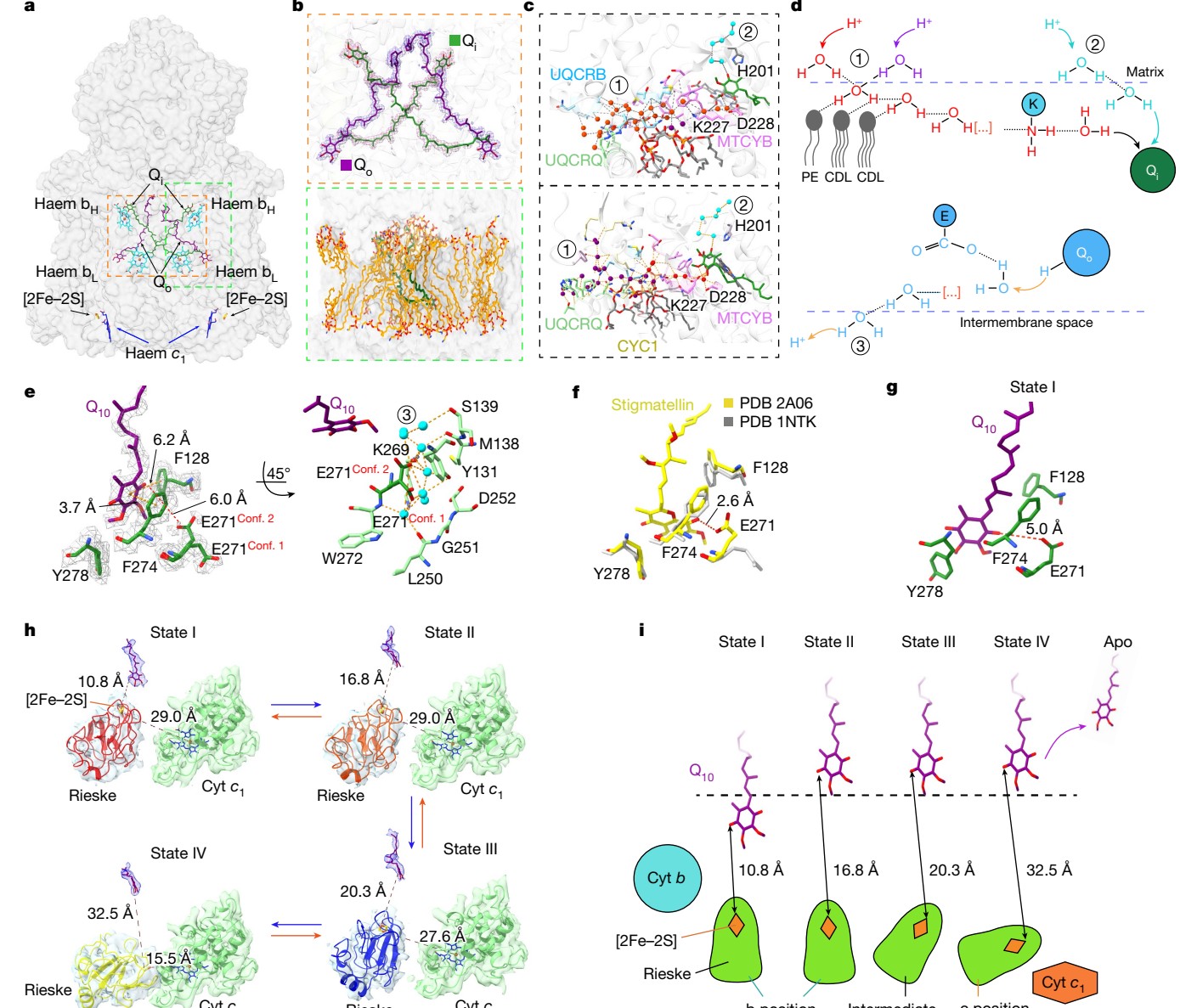

**Fig. 4 | Dynamic intermediates of CIII$_2$ revealed by high-resolution in situ cryo-EM. a**, The structures resolve all reactive centres. **b**, The endogenous Q$_{10}$ and lipid molecules in the CIII$_2$ Q-binding pocket and around the pocket entrance. These lipids are organized in a relatively ordered manner. **c**, Different views of the hydrogen-bonded network and the water chains for proton transfer. Red and purple spheres denote waters in the two branches of the bifurcated proton-influx path, whereas cyan spheres indicate waters in the single-wired proton-influx path. **d**, Schematic illustration of (1) bifurcated proton-influx path, (2) single-wired proton-influx path and (3) proton-outflow path. **e**, Structural details of the Q$_o$-binding site in the most abundant class. E271$^{MTCYB}$ displays dual conformations in this class. Q$_{10}$ and amino acids are

coloured purple and green, respectively. Blue spheres denote waters in the proton-outflow path near the Q$_o$ site. **f**, Comparison of E271 conformations between the apo-form CIII (PDB: 1NTK) and its complex with Q$_o$ inhibitor stigmatellin (PDB: 2A06) from the same view as shown in **e**. **g**, Structural details of the Q-binding site in state I with the closest distance to [2Fe–2S]. **h**, Four main Q-bound states of CIII as revealed by high-resolution in situ structures. The Rieske domain sequentially moves from b position to c position to shuttle electrons. This movement is coupled with the Q-binding state at the Q$_o$ site. Density maps for Q$_{10}$ headgroups are shown as blue mesh, whereas those for the Rieske domain and cyt c$_1$ are represented as transparent surfaces. **i**, Schematic representation of the coupling of ISP movement to Q-binding states.

single-particle cryo-EM and blue-native gel analysis[7,43,44]. However, details of the actual forms and conformations of these SCs within native membranes have remained elusive. The question of whether SCs offer enzymatic or functional advantages continues to be a subject of debate. Studies have not conclusively shown that SCs facilitate enhanced catalytic activity by directly channelling intermediate substrates[45,46]. A recent study suggested that SCs are non-essential for mouse bioenergetics and physiology[44], yet the authors maintained an open stance on their potential alternative roles, including even distribution of complexes throughout the membrane,

regulation of protein stability[47], minimization of reactive oxygen species production[48] and prevention of age-associated protein aggregation[49].

Our in situ structures indicate a distance of approximately 100 Å between the Q-entrances of CI and CIII$_2$, consistent with previous evidence[17,18]. The space between the two Q-entrances is filled with diffusive lipid molecules and seems to lack a confined space for direct substrate channelling in most of the SC structures we determined; the exception is type B, which seems to possess a semiclosed channel between CI and CIII$_2$ (Supplementary Fig. 9).

However, this is not the dominant form, indicating that the majority of $Q/QH_2$ is not necessarily shuttled between the CI and $CIII_2$ within the same SC.

Our study shows that the local geometry of the slightly curved membrane surrounding various forms of SCs fits the overall shape of the planar regions of mitochondrial cristae, consistent with previous findings that SCs predominantly localize in these regions[50–52]. The optimal spatial localization and organization of CI, $CIII_2$ and CIV can ensure a homogenous distribution of SCs within the inner membrane[53]. Notably, our current cryo-EM classification attempts did not detect noticeable cyt *c* density, indicating that binding of cyt *c* may occur instantly, whereas its diffusion within the intermembrane space may be the rate-limiting step. The homogenous arrangement of CI, $CIII_2$ and CIV helps to minimize the diffusion distance for $Q/QH_2$ and cyt *c*, potentially enhancing catalytic efficiency[54].

Despite a lack of direct Q-channelling structure between CI and $CIII_2$, we speculate that the variable local membrane curvature could regulate $Q/QH_2$ diffusion. Our structures show that the membrane near CI exhibits notable convex curvature towards the matrix, whereas it is distinctly concave near $CIII_2$, with a more planar region in between. Studies have demonstrated that membrane curvature can affect the lateral mobility and sorting of lipids, peptides and proteins, thereby influencing the behaviour of small molecules within the membrane[55,56]. Similar rules can be applied to the diffusion and sorting of $Q/QH_2$, potentially enriching the required substrates at the Q sites of CI and $CIII_2$.

Moreover, conventional approaches such as digitonin solubilization, blue-native gel analysis, cross-linking mass spectrometry and, particularly, mutagenesis on the basis of in vitro structures, may have limitations in their ability to accurately reflect the actual forms and physiological states of SCs in mitochondria. In the future, studies integrating functional and physiological analyses with the in situ imaging approach used in our current work will help to overcome these limitations and enhance our understanding of SC functions.

### A–D states and catalytic intermediates

In vitro purified mammalian CI consists of a mixture of the active and deactive resting states[45]. The closed structure, characterized by a reduced angle and tight interactions between the transmembrane and hydrophilic arms, corresponds to the active resting state, whereas the open structure is indicative of the deactive resting state. Consistently, our study also identified two main classes of CI, namely the active and deactive states, characterized by the closed and open conformations. At present, there are two opposing opinions about opening and closing of the two arms in the A–D transition: (1) that the open state is not linked to the catalytic cycle[45]; and (2) that the catalysis involves a succession of open and closed states, with the enzyme settling in these states upon cessation of catalysis[12].

In the fresh-condition batch, the active state predominated, with an increase in deactive states upon incubation at room temperature, indicating that the deactive state may arise with gradual depletion of substrates. In addition, active-apo and all different $Q/QH_2$-binding states were dominantly in the closed conformation (except state β) (Supplementary Fig. 11), implying that the closed form may be sufficient for the catalytic cycle. The global conformational changes between the A–D forms within the native mitochondrial membrane were considerably smaller than those observed in detergent-purified[12] and membrane scaffold protein nanodisc reconstituted respiratory CI[28], indicating that the energetic cost required for the conformational changes in the native lipid bilayer environment may be higher than those under in vitro conditions (Extended Data Fig. 10). Notably, our extensive 3D classification of in situ images did not yield a class that resembled the reported relocation of the helix ND6-THM4 (Supplementary Fig. 13)[12]. We speculate that such a relocation may be energetically unfavourable, as this would require substantial reorganization of surrounding lipids associated with CI.

## In situ imaging beyond mitochondria

Our study used in situ cryo-EM imaging techniques to directly visualize SC structures within mitochondria. This approach bypasses the limitations associated with traditional in vitro purification methods, which could result in loss of native ligands and physiological states, leading to compositional and conformational artefacts. Our approach preserves both the native membrane environment and the electrochemical proton gradient. This enabled concurrent analysis of SC structures in reaction at both atomic detail and large-scale organization, achieving sub-2 Å local resolution. Optimized in silico classification of the in situ cryo-EM data enabled us to capture a series of dynamic ligand-binding states in the respiratory chain, eliminating the requirement for analogues or reaction inhibitors commonly used in conventional methods and thus providing new insight into the mechanisms of mitochondrial SCs that reflect physiological behaviour. Notably, the preservation of SC structures in native membranes enabled direct visualization of numerous surrounding lipids, which have pivotal roles at the interaction interfaces among the complexes as well as within the Q-binding pockets. These findings have potential extended applications in studies to elucidate the effects of many analogues, drugs and inhibitors on mammalian mitochondria, shedding light on SC behaviours under various physiological and pathological conditions, including cardiopathy, diabetes, neurodegeneration and cancer. Furthermore, our in situ approach could be extended to structural studies of native membrane protein complexes within various organelles, such as cilia, chloroplasts, Golgi and lysosomes.

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

## Methods

### Preparation of porcine mitochondria and cryo-EM grids

Mitochondria were isolated from porcine hearts following a modified version of a protocol originally described by A. L. Smith[57]. Before mitochondrial extraction, the pig hearts were subjected to three distinct treatment conditions: (1) fresh—immediately placed on ice for all subsequent procedures; (2) mild—incubated at room temperature for 40 min and put on ice to quickly cool down for isolation; and (3) harsh—incubated at room temperature for more than 4 h before being cooled on ice. The isolated mitochondria were then resuspended in a solution containing 0.25 M sucrose, 10 mM Tris-buffered with $H_2SO_4$ and 0.2 mM EDTA at pH 7.8. The suspension was adjusted to achieve a final optical density at 600 nm of 1.3 absorbance units.

For cryo-EM grid preparation, 3.3 μl of the mitochondrial suspension was applied to each Quantifoil holey carbon grid (R2/1, 300 mesh gold). Grids were incubated for 5 s in a Vitrobot Mark IV (Thermo Fisher Scientific) chamber maintained at 8 °C and 95% relative humidity. Excess solution was blotted using standard Vitrobot filter paper before the grids were rapidly plunged into liquid ethane at a temperature of approximately −170 °C.

### Cryo-ET data collection

Grids were initially screened for optimal ice conditions using a 200 kV Glacios microscope (Thermo Fisher Scientific) at the Yale Science Hill Electron Microscopy Facility. Selected grids were subsequently transferred to a 300 kV Titan Krios microscope (Thermo Fisher Scientific), equipped with a Bioquantum Energy Filter and a K3 direct electron detector (Gatan), for high-resolution data acquisition at the Yale West Campus Electron Microscopy Facility. Automated data collection was facilitated using SerialEM software[58] and Gatan DigitalMicrograph. All images were captured in superresolution mode, with a physical pixel size of 6.1 Å (effectively 3.05 Å in superresolution). A total of eight tilt series were collected, targeting a relatively high defocus range, from −6 μm to −10 μm, for better contrast to guarantee a more reliable initial reconstruction. A grouped dose-symmetric scheme, spanning from −60° to 60° at 2° increments, was used for tilt series acquisition, with an accumulated dose of 100 $e^-/Å^2$.

### Cryo-ET reconstruction and subtomogram averaging

Tomogram reconstruction was streamlined using custom scripts. Initial frame alignment was performed using MotionCorr2 (ref. 59), followed by micrograph binning at a factor of two. Tilt series stacks were generated using in-house scripts. All tilt series were aligned and reconstructed using AreTomo 1.2.5 (ref. 60). Initial contrast transfer function (CTF) parameters were estimated with GCTF[61] and cryoSPARC[62]. Raw micrographs and reconstructed results were visualized and diagnosed using IMOD[63] and ChimeraX[64].

Individual SC particles were picked in EMAN2 (ref. 65). Metadata preparation yielded 12,000 subtomogram particles in RELION-4.0 (ref. 66) with a binning factor of 2 (pixel size 12.2 Å). Following two rounds of 3D classification, 806 SC particles were selected for final refinement, resulting in a 37 Å subtomogram averaging map. Resolution was assessed using Fourier shell correlation with a threshold of 0.143 in RELION-4.0 (ref. 66). The averaged map was backprojected onto the original tomogram using the subtomo2Chimera code, available at https://github.com/builab/subtomo2Chimera.

### Single-particle cryo-EM data collection

Automated data acquisition was performed using either a Glacios or a Titan Krios electron microscope (Thermo Fisher Scientific). The Glacios was equipped with a K3 direct electron detector (Gatan) and operated at 200 kV at a pixel size of 0.434 Å in superresolution mode, with an objective aperture of 100 μm. The Titan Krios, also equipped with a K3 direct electron detector, was operated at 300 kV at a pixel size of 0.416 Å in superresolution mode with a Gatan energy filter. Automatic data collection was facilitated using the SerialEM software package[58]. Multishot acquisition parameters were set at 3 × 3 holes per imaging location, with four exposures per hole at 200 kV and five exposures per hole at 300 kV. The total electron dose was fractionated to 42 $e^-/Å^2$ for the Glacios and 50 $e^-/Å^2$ for the Titan Krios, distributed across 45 frames at 40 ms per frame. Defocus parameters ranged from −1.0 μm to −3.0 μm for the 200 kV dataset and from −1.3 μm to −3.0 μm for the 300 kV datasets. Details of the data collection are summarized in Supplementary Tables 1–6.

### Preprocessing

For all datasets, motion correction was performed using MotionCor2 (ref. 59) or cryoSPARC[62]. The CTF of each motion-corrected micrograph was estimated using Gctf[61] or cryoSPARC[62]. Particles were picked with Gautomatch or cryoSPARC using an iterative sorting strategy as described below. Cryo-EM scripts used for real-time data transfer and on-the-fly preprocessing can be downloaded from https://github.com/JackZhang-Lab.

### Overall particle selection and sorting strategy

Owing to the challenges posed by low signal-to-noise ratios and a highly congested macromolecular environment (Extended Data Fig. 1a), traditional particle selection methodologies were insufficient for generating datasets amenable to reliable two-dimensional (2D) classification, ab initio three-dimensional (3D) reconstruction and subsequent local refinement. To address this issue, we implemented an iterative strategy to optimize particle selection and sorting. The approach involved several rounds of iterative 2D particle picking, 2D classification and 3D analyses including ab initio 3D reconstruction, 3D classification and multilevel local refinement. Unlike the conventional particle selection approach, our strategy used Gautomatch and cryoSPARC[62] for template matching to gradually increase the resolution of 3D projections as the reconstructions were progressively improved over cycles. We used several independent sources of references to cross-validate the final results. To maximize the yield of high-quality particles, particles from the classes that show clear features of SCs in all cycles were merged for subsequent 3D cross-classification. More details of the strategy are explained in the following sections.

### Initial 3D reconstruction with surrounding membranes (type A)

Conventional 2D classification failed to generate meaningful class averages using images selected from our in situ cryo-EM micrographs of mitochondria for three main reasons: (1) thick samples that led to low signal-to-noise ratios and large defocus variations, (2) a crowded environment that affected particle detection and alignment, and (3) strong membrane signals that dominated the alignment, leading to blurred averages of protein regions (Extended Data Fig. 1b).

To address this, we initially used the strong membrane signals and focused on the side views surrounded by membranes using 2D classification. These side views in principle contained sufficient orientational information for a complete 3D reconstruction. At the outset, protein signals were completely averaged out in the 2D classification, whereas the membranes were well aligned owing to the strong side-view signals (Extended Data Fig. 1c). We then conducted several cycles of 2D classification to focus only on particles exhibiting clear membrane signals.

Through comprehensive 2D analyses, we found that regions potentially harbouring mitochondrial SCs exhibited special features of local curvature. Specifically, these regions were characterized by membrane signals that seemed to be concave towards the matrix direction, indicative of the presence of $CIII_2$ (Extended Data Fig. 1c–e). By merging particles from classes with characteristic concave membranes surrounding $CIII_2$ and conducting further 2D classification, we achieved improved 2D averages showing clear membrane features around $CIII_2$ (Extended Data Fig. 1d). Notably, extra protein densities adjacent to $CIII_2$ were obvious,

probably representing CI or CIV densities. However, it was unclear how many types of respiratory SC exist in native mitochondria and whether CI, CIII$_2$ and CIV always appear in the form of SCs or just partially.

To further address these observations and obtain unbiased density maps, we used four independent methods to generate initial references: (1) cryo-ET subvolume averaging (Supplementary Fig. 1), (2) ab initio reconstruction using particles assigned to the 2D averages with visible protein densities (Extended Data Fig. 1f) and characteristic CIII$_2$ membrane features (Extended Data Fig. 1g), (3) ab initio 3D reconstruction using particles after membrane signal subtraction (Extended Data Fig. 1h), and (4) models generated from random selection of unsorted particles or random noise (Extended Data Fig. 1i). All these references were combined for 3D classification and subsequently used for local refinement and focused classification (Extended Data Fig. 1i). Given that in situ cryo-EM datasets are more heterogeneous than conventional single-particle datasets, we included 'false references' generated from approach (4) for better classification. Finally, particles corresponding to classes showing clear features of type-A SC were re-extracted and merged for further classification and refinement (Extended Data Fig. 1i).

## Cross-classification of multiple SCs
Around the reconstructed SC I$_1$III$_2$IV$_1$ (type A) map, we observed extra densities, clearly indicating that more proteins bound to the type-A SC to form larger SCs. We suspected that more types of SC existed in native mitochondrial membranes. Preliminary results from both large single-particle 3D classification at low resolutions and cryo-ET subvolume averaging confirmed this speculation. To further improve the accuracy of 3D classification for high-resolution refinement, we deliberately provided extra false references generated from random subdatasets using discarded particles from previous cycles. These false references served to randomly absorb low-quality and falsely picked particles, leading to a relatively clean dataset for the target class. We then accumulated particles classified into good classes, defined by clear secondary structures, over several cycles. Owing to the crowded mitochondrial environment, misclassified and misaligned particles were always present. To address this, we reorganized the particles by merging those that fell into classes generating similar 3D maps. We selected multiple references from different classes, including those considered 'bad' and reperformed 3D classification on each subdataset. Afterwards, we recombined all subsets of different classes that were considered 'good' and reclassified them. On the basis of these results, we then merged all the particles belonging to a specific target from previous cycles and performed a further cycle of 3D classification on the merged dataset. This further classification used high-resolution references generated from previous classification cycles and local refinement to discard low-quality or misclassified particles.

After numerous rounds of cross-classification followed by local refinement, we identified various other types of SC, including the three other main classes: type B (I$_1$III$_2$IV$_2$), type O (I$_2$III$_2$IV$_2$) and type X (I$_2$III$_4$IV$_2$). In addition to the four main classes, other classes such as I$_1$III$_1$, I$_4$III$_4$IV$_4$ and even higher-order assemblies were observed; however, they were not subject to further refinement in this study owing to the low population. Subsequently, we cross-validated our classification results by providing a set of references lacking the correct form of the SC for subclassification of each class. We also performed further reference-free 2D classification after 3D classification and refinement to verify different forms of SC. This allowed us to visualize the distinct features of the four main classes from 2D averages directly, without imposing any references. Only datasets converging to the correct form of supercomplex, regardless of the initial references used, were included in the final multilevel local refinement and focused 3D classification.

## Multilevel local refinement and focused 3D classification
A hierarchical masking strategy was used for local refinement on all four main types of SC. Specifically, the mask size was incrementally reduced to focus on distinct regions of each type of respiratory SC, ensuring stable local refinement. We partitioned the type-A SC into five principal domains: (1) CI hydrophilic region, (2) CI hydrophobic region, (3) CIII$_2$, (4) CIV and (5) lipid environment.

Before the multilevel local refinement, the type-A SC was refined to 3.39 Å overall using images binned two times (1.664 Å per pixel after binning) with 1,113,902 high-quality particles. This included type B, type O and type X, as they all share the type-A region. We recentred and re-extracted these particles, generating 1,050,463 final particles for subsequent local refinement (particles near the edges were excluded after re-extraction). Initially, the resolutions of CI, CIII$_2$ and CIV worsened slightly (approximately 3.5 Å) after the first cycle of refinement using the unbinned particles (0.832 Å per pixel). Further improvement was achieved by optimizing several local refinement parameters, including optimization of mask sizes, global CTF, local CTF refinement, local angular refinement and non-uniform refinement[67].

By iteratively applying these techniques, we refined the maps of the hydrophilic region of CI and the hydrophobic regions of CI, CIII$_2$ and CIV to average resolutions of 2.46 Å, 2.58 Å, 2.31 Å and 2.66 Å, respectively (Supplementary Fig. 2 and Supplementary Tables 1–6). Even smaller regional masks, focused on CI and CIII$_2$, further improved local resolutions. Local resolutions in most of the protein regions of CIII$_2$ ranged from 1.8 to 2.4 Å (Supplementary Fig. 2c). Focused classification and refinement for specific subdomains, such as the Q/QH$_2$ binding sites, yielded further improvements that aided in model building. For more complex regions, such as the lipid environment surrounding the transmembrane regions of the SCs and Q/QH$_2$ binding sites, further levels of focused classification and local refinement were performed. To ensure seamless integration of adjacent regions, all local masks were manually created so that pairs of adjacent masks contained sufficiently large areas for the generation of final composite maps using the smaller regions individually refined. All locally refined segments were integrated into a composite map in ChimeraX[64].

Similar multilevel refinement approaches were used to determine the structures of other forms of respiratory SC. Detailed parameters and refinement results are summarized in Supplementary Figs. 2–5 and Supplementary Tables 1–6.

## Membrane signal detection and weakening
One of the critical bottlenecks limiting high-resolution cryo-EM reconstruction of membrane proteins in their native environment is the severe signal interference from surrounding membranes. This interference can significantly affect several steps in cryo-EM data analysis, including ab initio reconstruction, Euler angle determination, and 2D and 3D classification, as well as refinement of alignment parameters. To address this issue, we developed a computational toolkit to detect membrane signals from 2D averages, estimate the local geometry of detected membranes, and suppress or remove these signals to substantially improve the alignment reliability of mitochondrial complexes in native membrane environments.

Initially, we generated a series of 15–30 computationally simulated 2D projections of lipid bilayers, with local curvatures ranging from 0 nm$^{-1}$ to 0.02 nm$^{-1}$. These simulated 2D membranes served as templates for detection of the side-view signals of mitochondrial membranes using Gautomatch. Subsequently, three to five cycles of 2D classification were performed to discard low-quality and non-membrane particles, resulting in a subset of particles showing clear side views of lipid bilayers. We then estimated the approximate orientation and centre of each individual lipid bilayer on the basis of its corresponding 2D average using the Radon transform. Local curvature was determined by maximizing the cross-correlation between each 2D average and a series of simulated lipid bilayers. These curves were rotated and translated using alignment parameters from 2D classification generated by cryoSPARC[62]. Centres of each membrane segment were refined by maximizing the normalized cross-correlation between the raw image and transformed

2D average. Using these estimated parameters, we approximated the principal signals of each membrane segment by locally averaging the image intensities along the membrane curve within a soft mask, which was around 25% larger than the typical lipid bilayer we estimated. Membrane signals that had dominated the alignment in the raw images were weakened to enhance protein signal contributions for subsequent reconstruction, alignment, classification and local refinement. This improved the signal contributions from protein regions for the initial alignment, akin to the critical effects observed in our previously described microtubule signal subtraction method[68,69]. Finally, alignment and classification parameters were applied to the raw images along with membrane signals for subsequent local refinement and focused classification.

## Membrane modelling and geometry analysis

The in situ mitochondrial respiratory chain complexes largely preserved the native state of the membrane architecture, as evidenced by exceptionally clear density maps (Extended Data Fig. 2) compared with previously published in vitro structures. This high fidelity in density was observable in both the final 3D reconstructions and the post-3D-refinement 2D class averages, enabling direct modelling of native membrane structures.

The model building for the inner membrane structures surrounding the mitochondrial SCs involved a four-step procedure. First, discrete points were sampled from the raw signals in a given density map—such as the type-A SC—on the basis of binarized membrane density. A 2D plane was fitted by least-square minimization; the normal vector of each SC was estimated and the coordinate system was rotated so that this vector aligned with the $z$ axis. Second, these sampled discrete points were used to generate two smooth, curved surfaces with a thickness of around 4 nm. Third, planar phospholipid bilayer structures were generated to match the geometry of these estimated surfaces. Finally, the information from the second and third steps was integrated to geometrically deform each planar membrane structure into a smooth, curved surface.

To optimize the initial sampling for membrane model building, we categorized the membrane structures surrounding the protein into three distinct groups: structured lipids, surface-associated lipids and generic bilayer lipids. The first category, structured lipids, included lipids that are closely associated with the transmembrane regions of the protein. This close association enabled identification and direct atomic-level modelling of these specific lipid species, which have also been observed in previously reported structures purified using detergent. The second category, surface-associated lipids, comprised lipids situated around the immediate periphery of the protein, forming a pseudo-lattice structure. Within this lattice, partial phosphatidyl head groups and hydrophobic tails could be discerned. Our in situ density maps allowed us to unambiguously determine the locations of individual lipids in this category; however, the current quality of the density maps does not permit identification of the specific types of lipid present. The third category, generic bilayer lipids, represented a region farther from the protein where only the density features corresponding to the bilayers could be observed. We used a generic phospholipid membrane model to approximate the probable horizontal positions of the phosphatidyl headgroups. Owing to the fluid nature of the lipid bilayer and the high level of noise in the density maps, the central positions of these generic bilayer lipids may still vary among different subclasses even after focused classification. However, the average geometric features and the central locations of the membranes were notably consistent across each of the four main types of SC. Therefore, these generic bilayer lipids were used solely for calibrating the central locations and orientations of the phospholipid bilayer, rather than representing the actual positions of individual phospholipid molecules within the bilayer of each SC. This approach facilitated analysis of the overall geometric changes among the SCs, albeit not at the level of individual phospholipid molecule structures.

To achieve a sufficiently smooth model for the generic bilayer lipids, we performed real-space refinement of the initial structures using the Coot software[70]. The refined structures were subjected to further smoothing using a local Gaussian filter to minimize residual noise in localized membrane regions. This step enabled precise estimation of the contour map and the local curvature at each point (Fig. 2b). We used the CHARM-GUI web service[71] to generate a simulated rectangular planar phospholipid bilayer. This planar structure was then mapped on to the curved surfaces that were obtained after Gaussian smoothing. This mapping process yielded a curved membrane model that optimally fit the density map. From these estimated surfaces, information about the local geometry of the membranes surrounding the mitochondrial SCs could be directly retrieved for subsequent geometry analyses and comparisons.

## Model building, refinement and validation

The atomic models were built manually using Coot[72]. First, high-resolution structures of bovine CI (PDB: 7QSK), bovine CIII$_2$ (PDB: 2A06) and bovine CIV (PDB: 5XDQ) were fitted into the corresponding map as a rigid body using ChimeraX[64]. Then, the fitted model was manually mutated, adjusted and real-space refined to correct errors in local regions to best match the density maps using Coot[72]. The final model was refined using phenix.real_space_refine[73] with geometric constraints and validated using MolProbity[74]. Figures were generated using UCSF ChimeraX[64] and PyMOL.

## Reporting summary

Further information on research design is available in the Nature Portfolio Reporting Summary linked to this article.

## Data availability

All cryo-EM maps and atomic coordinates for mitochondrial SCs have been deposited in the Electron Microscopy Data Bank (EMDB) and the Protein Data Bank (PDB), including entire and locally refined maps of type-A SCs under accession codes EMD-42231/PDB 8UGP, EMD-42225/PDB 8UGH, EMD-42226/PDB 8UGI; type-B SCs under EMD-42227/PDB 8UGJ; type-O SCs under EMD-42230/PDB 8UGN; and type-X SCs under EMD-42233/PDB 8UGR. Different states of CI with Q$_{10}$ are deposited under EMD-42165/PDB 8UEO, EMD-42166/PDB 8UEP, EMD-42167/PDB 8UEQ, EMD-42168/PDB 8UER, EMD-42176/PDB 8UEZ; different CI deactive classes under EMD-42169/PDB 8UES, EMD-42170/PDB 8UET, EMD-42171/PDB 8UEU, EMD-42172/PDB 8UEV, EMD-42173/PDB 8UEW, EMD-42174/PDB 8UEX, EMD-42175/PDB 8UEY; different states of CIII$_2$ under EMD-42221/PDB 8UGD, EMD-42222/PDB 8UGE, EMD-42223/PDB 8UGF, EMD-42224/PDB 8UGG; and high-resolution structures representative of the complex under EMD-42143/PDB 8UD1, EMD-42228/PDB 8UGK, EMD-42229/PDB 8UGL.

## Code availability

All codes involved in general cryo-EM/ET data processing and structure determination of membrane proteins will be publicly available at https://github.com/JackZhang-Lab.

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

**Acknowledgements** All cryo-EM datasets were collected at the Yale CryoEM Resource facilities (Science Hill and West Campus), funded in part by the National Institutes of Health (NIH) grant S10OD023603 awarded to F. Sigworth and Brookhaven National Laboratory. We thank S. Wu, J. Lin and K. Zhou at Yale University and L. Wang, J. Kaminsky and G. Hu at the Laboratory for BioMolecular Structure for technical support with electron microscopy, as well as M. Gu, B. Ma and Z. Huang for assistance in preparing materials necessary for the manuscript. This work was supported by startup funds from Yale University and by the NIH grant R35GM142959 awarded to K.Z.

**Author contributions** K.Z. and J.Z. designed the project. W.Z. prepared all samples for cryo-EM and cryo-ET studies. W.Z. and K.Z. collected all cryo-EM and cryo-ET data. All authors contributed to cryo-EM and cryo-ET data processing, built the atomic models, analysed the results and prepared the manuscript.

**Competing interests** The authors declare no competing interests.

**Additional information**
**Correspondence and requests for materials** should be addressed to Jiapeng Zhu or Kai Zhang.

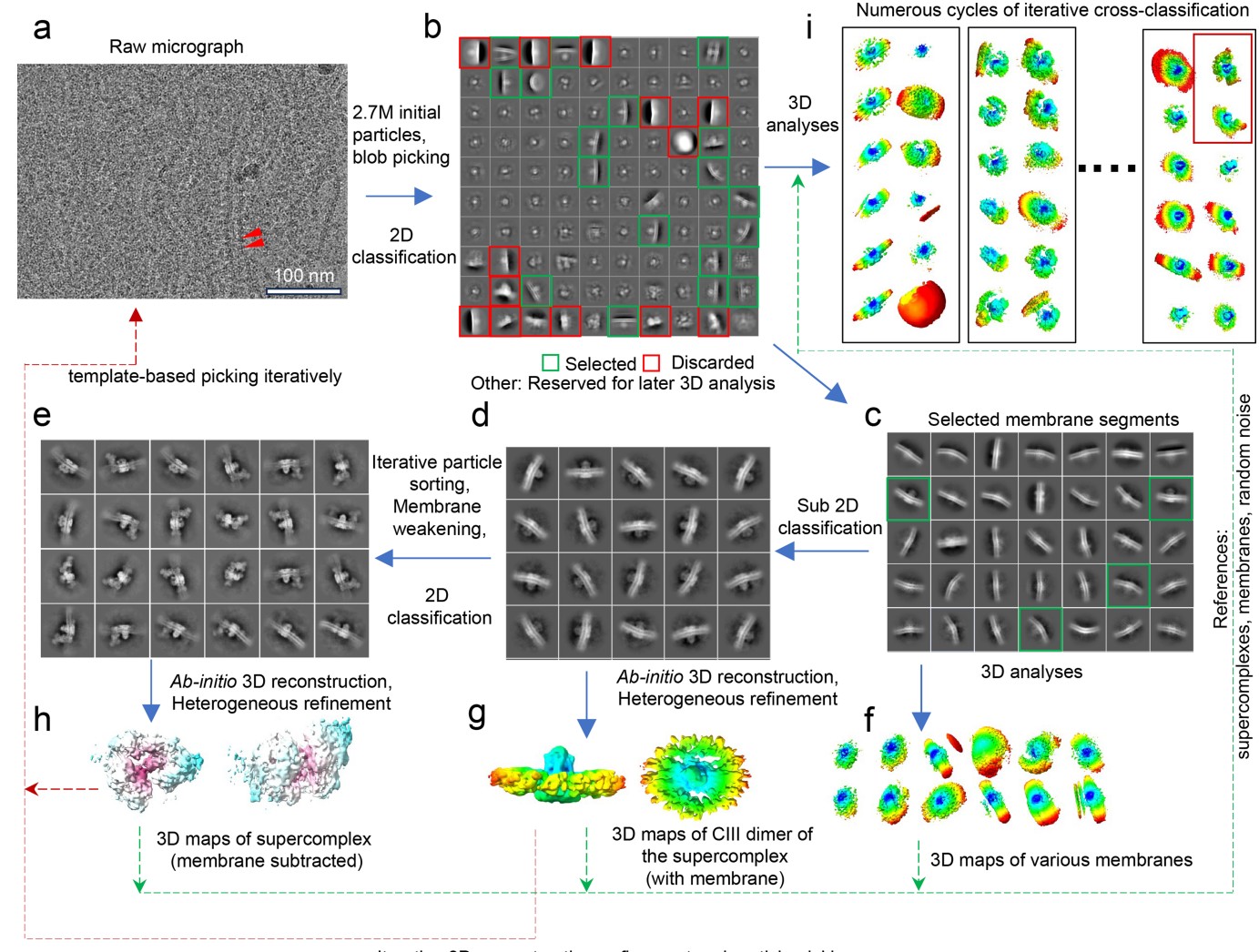

**a** Raw micrograph

**b** 2.7M initial particles, blob picking / 2D classification

☐ Selected  ☐ Discarded
Other: Reserved for later 3D analysis

template-based picking iteratively

**i** Numerous cycles of iterative cross-classification

3D analyses

**c** Selected membrane segments

Sub 2D classification

3D analyses

References: supercomplexes, membranes, random noise

**d** *Ab-initio* 3D reconstruction, Heterogeneous refinement

Iterative particle sorting, Membrane weakening, 2D classification

**e**

**h** 3D maps of supercomplex (membrane subtracted)

**g** *Ab-initio* 3D reconstruction, Heterogeneous refinement / 3D maps of CIII dimer of the supercomplex (with membrane)

**f** 3D maps of various membranes

Iterative 3D reconstruction, refinement and particle picking
(Many cycles until converged)

**Extended Data Fig. 1 | Schematic Overview of Initial Single-Particle Cryo-EM Data Analysis Workflow. a**, High-magnification micrograph exemplifying a typical single-particle cryo-EM sample. **b**, Preliminary results of reference-free 2D classification. **c**, Representative outcomes of 2D sub-classification focused on particles displaying discernible side views of mitochondrial membranes. **d**, Illustrative 2D classification of mitochondrial supercomplex particles derived from the prior 2D classes; features reveal distinct concave membrane morphologies in the CIII$_2$ regions. **e**, Exemplary 2D class averages after extensive particle sorting and membrane signal weakening. **f-h**, 3D reconstructions generated using particles selected from steps (**c**), (**d**), and (**e**), respectively. **i**, Typical results of the iterative cross-3D classification. Reference maps and particle classification/alignment parameters were progressively refined over multiple cycles. Upon reaching this stage, reliable 3D reference models were obtained for subsequent multi-level refinement and focused classification. Particles classified into SCs were merged for all ensuing data processing steps.

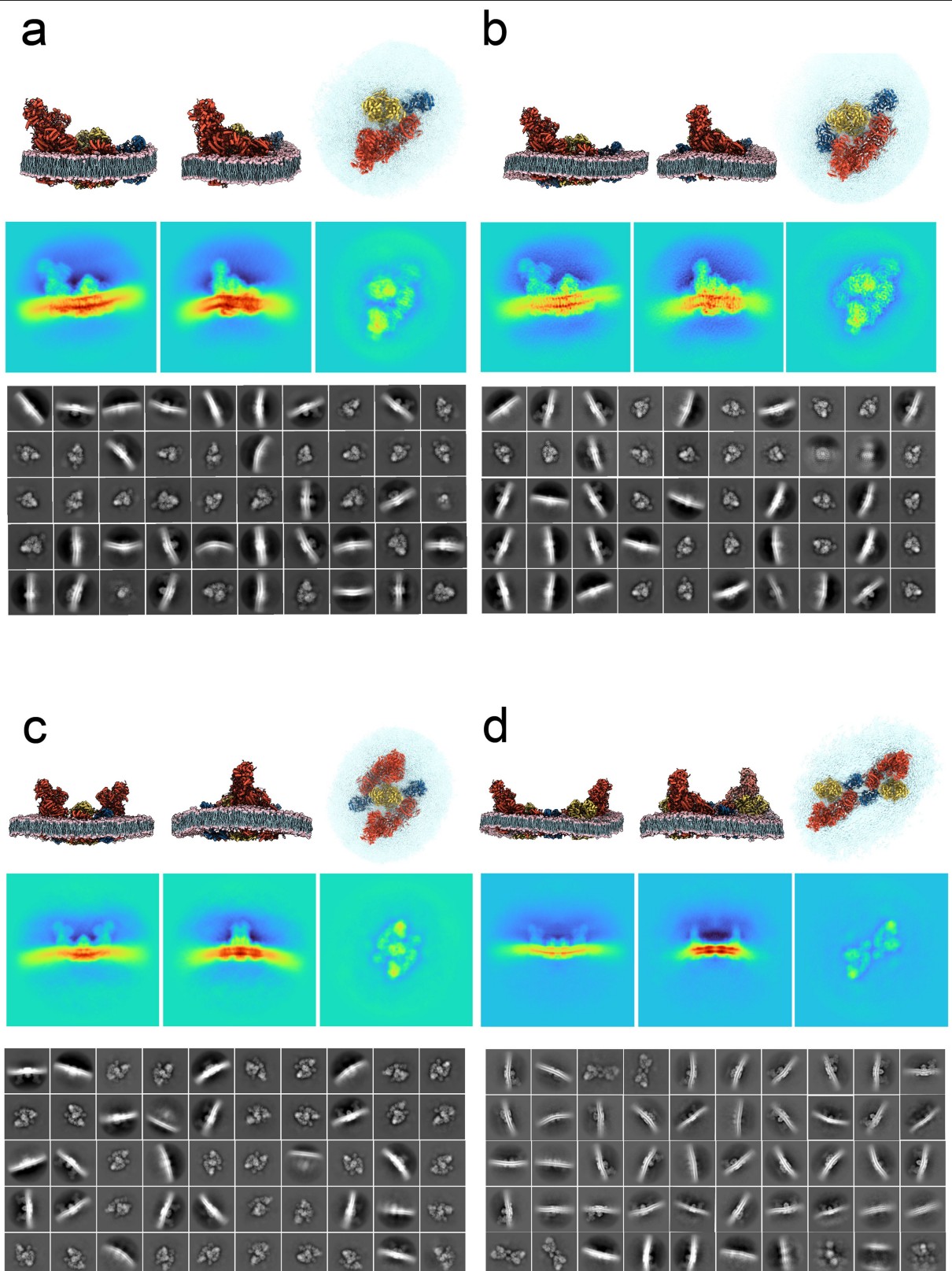

**Extended Data Fig. 2 | Evaluation of the 3D classification by post-3D-refinement 2D classification. a-d**, Atomic models (top) from three representative views, the corresponding projections from the same orientations (middle) and representative reference-free 2D class averages of type-A (a), -B (b), -O (c) and -X (d) SCs after 3D classification and refinement.

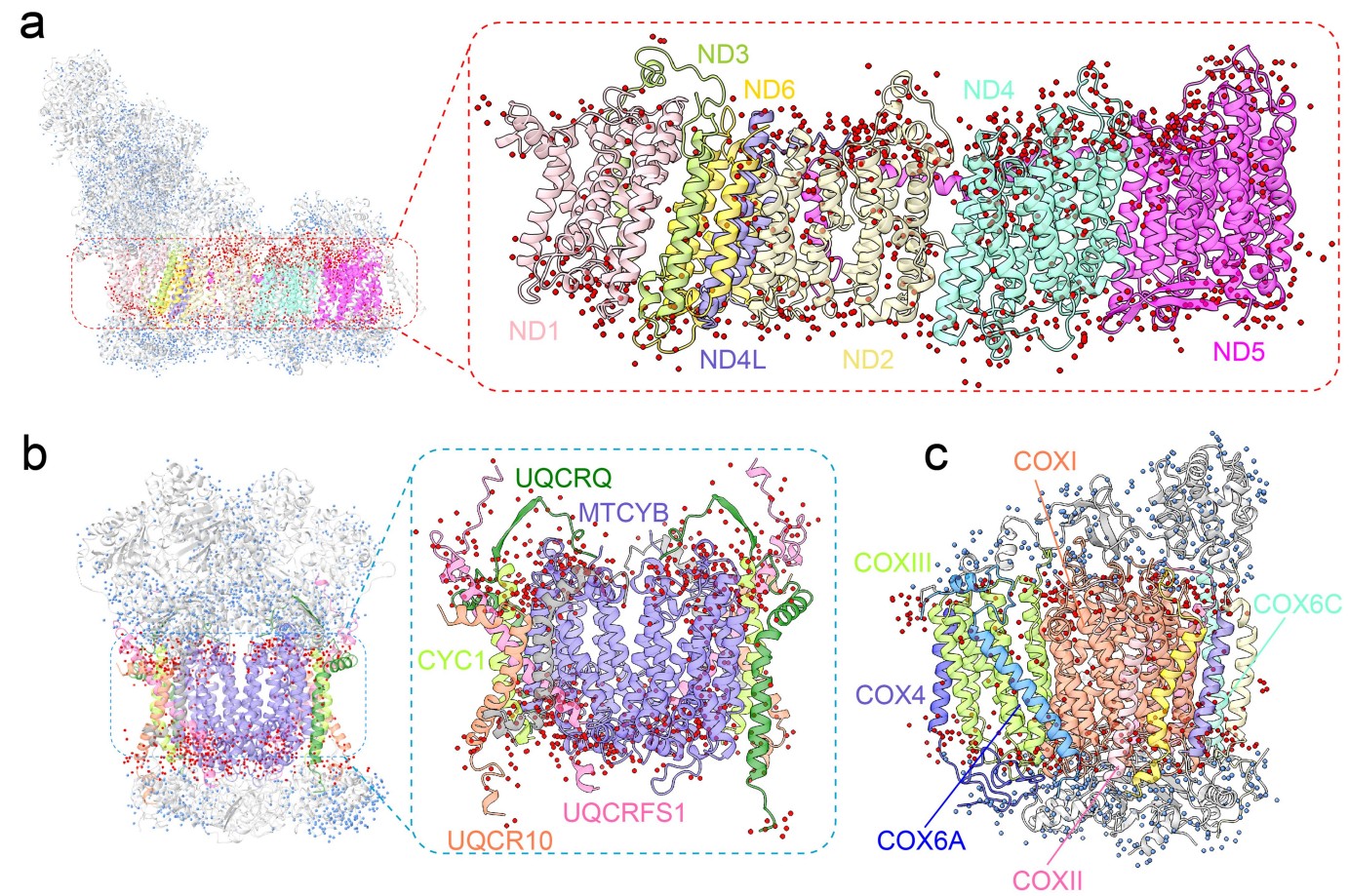

**Extended Data Fig. 3 | Water molecules in CI, CIII₂, and CIV. a-c**, The water molecules bound in CI, CIII₂ and CIV. Blue dots represent water molecules in the hydrophilic regions, while red dots indicate water molecules near the hydrophobic regions. The core subunits of CI, CIII₂ and CIV are color-coded as shown in the figure.

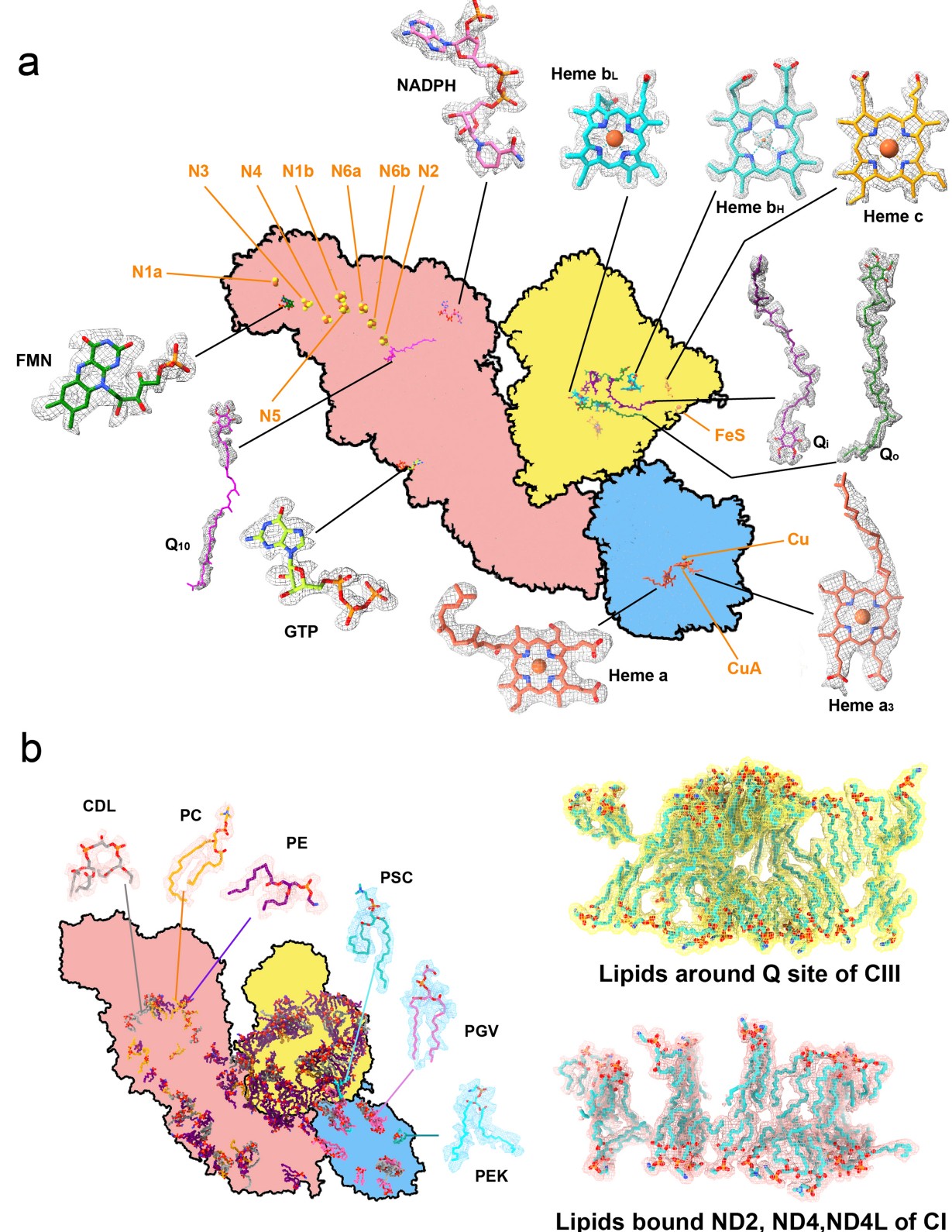

**Extended Data Fig. 4 | Endogenous cofactors and representative structured lipids revealed by the high resolution *in-situ* cryo-EM structures. a**, Representative density maps of endogenous cofactors and atomic models fitted. The cartoon in the center represents the contour of the type-A supercomplex projection with the ligand position assigned. **b**, Structured and associated lipids identified from the high-resolution cryo-EM density of type-A supercomplex.

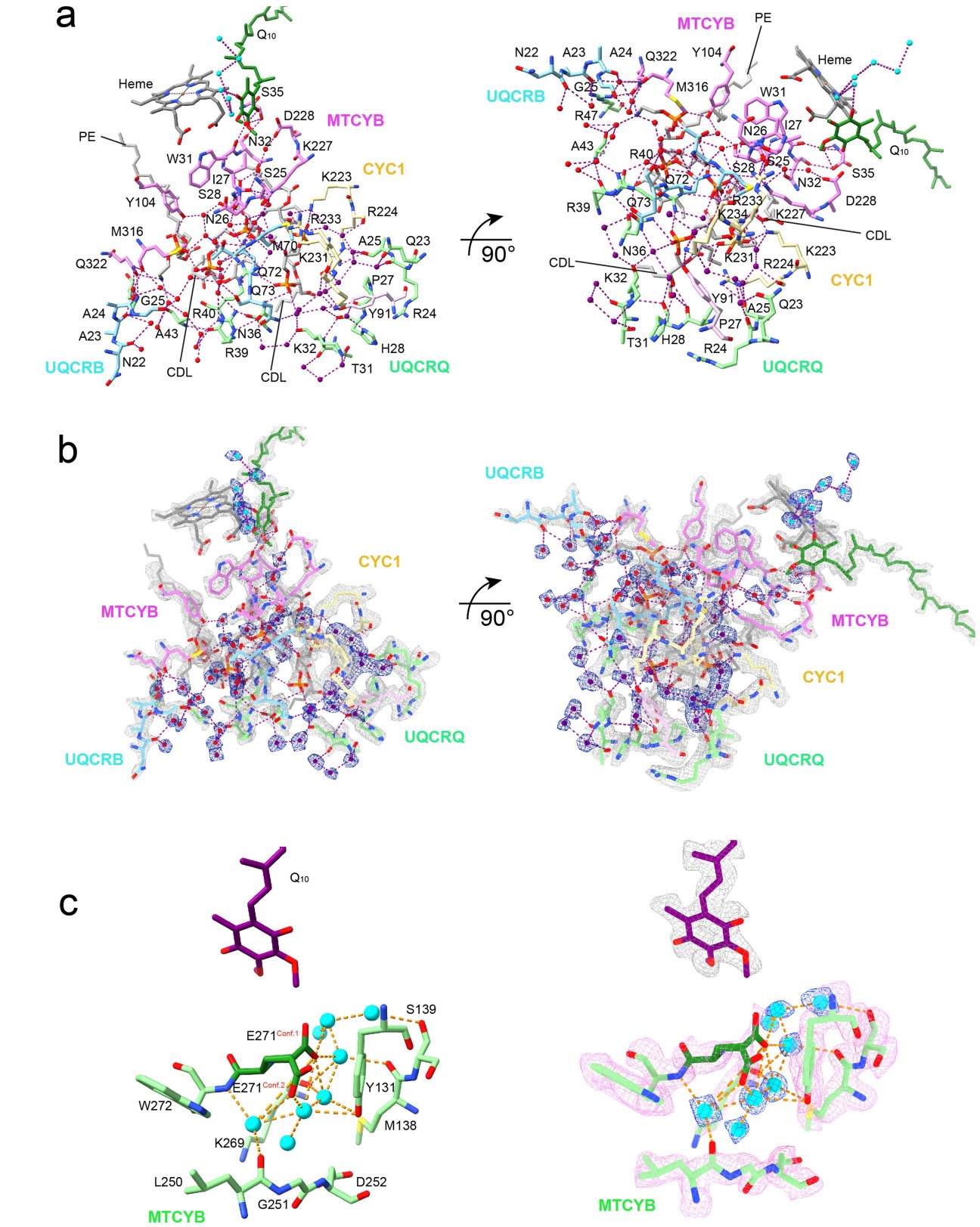

**Extended Data Fig. 5 | Complex hydrogen-bonded networks near the $Q_i$ sites for proton transfer in CIII$_2$. a**, These networks consist mainly of water molecules, supported by their interacting residues, and polar headgroups of lipids, forming the proton uptake path near the $Q_i$ site and the proton release path near the $Q_i$ site. **b**, The high-resolution map enables us to confidently build water molecules in $Q_i$ site and their surrounding residues and lipids that form these intricate hydrogen-bonded networks. **c**, The proton-transfer water chain and local density map near the $Q_o$ Site.

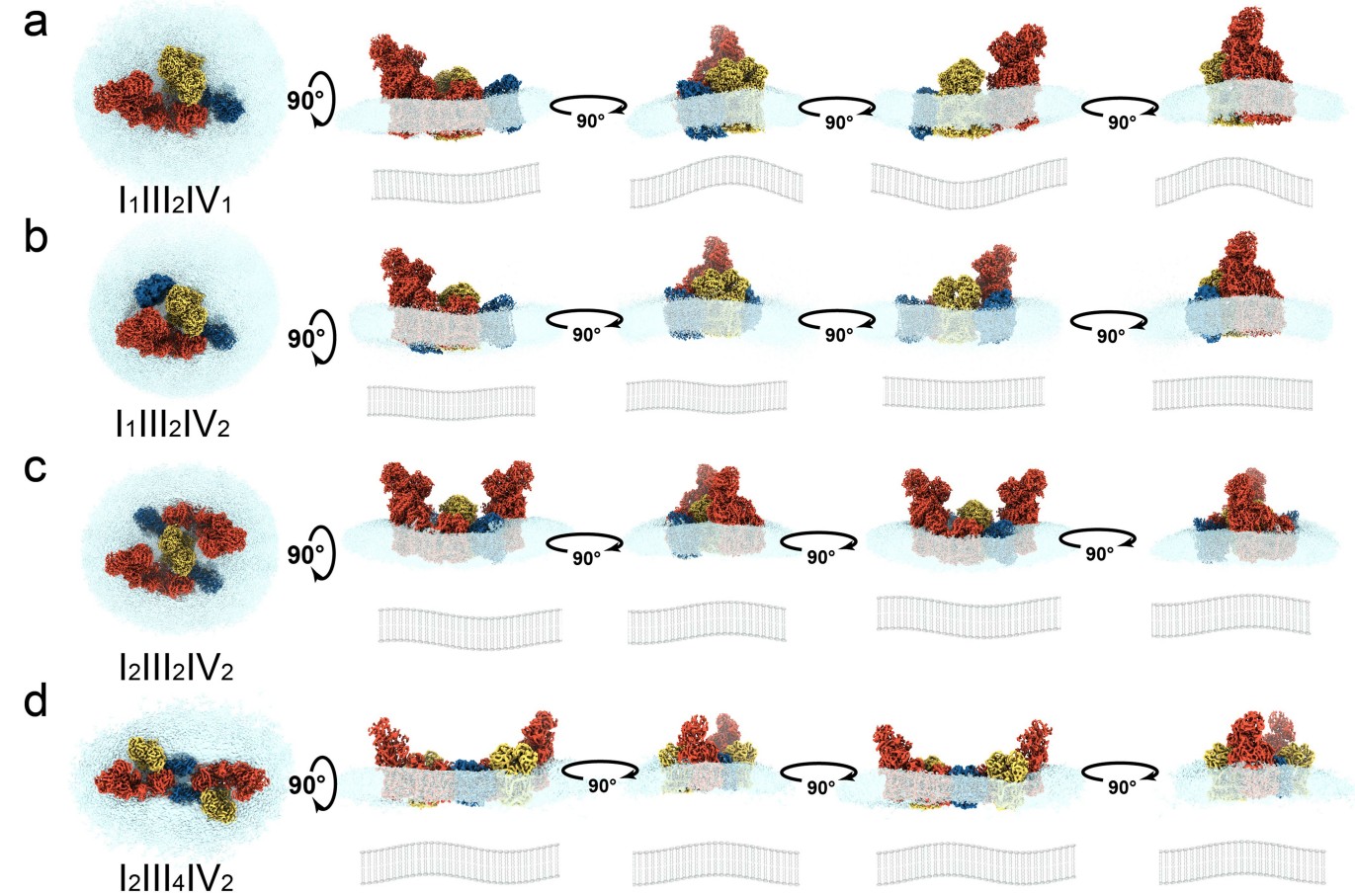

**a** $I_1III_2IV_1$

**b** $I_1III_2IV_2$

**c** $I_2III_2IV_2$

**d** $I_2III_4IV_2$

**Extended Data Fig. 6 | Influence of supercomplexes on surrounding membrane curvature. a-d**, Side views illustrating the impact of SCs on membrane curvature.

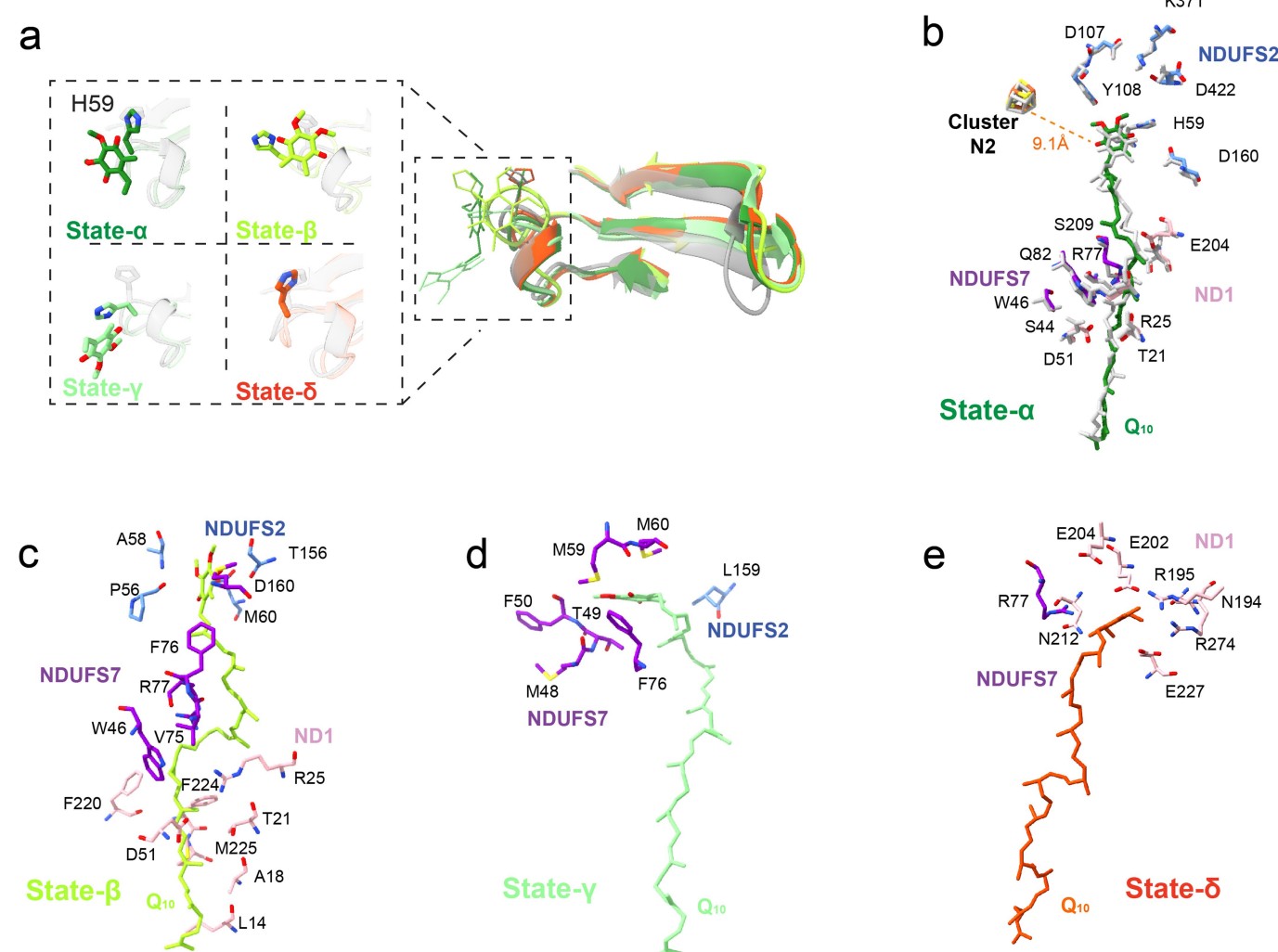

**Extended Data Fig. 7 | Different Q-binding states and conformational changes in the Q-channel. a**, Superpositions of the four different Q-binding states with the active-apo state (grey) reveal not only conformational alterations in the $Q_{10}$ headgroup and H59$^{NDUFS2}$ (left panel) but also significant changes in long-range structures away from $Q_{10}$ headgroup (right panel).

**b-e**, Superimposition of the porcine *in-situ* supercomplex structure with the bovine CI (grey) incorporated into MSP nanodisks indicates that our fully occupied state resembles the active form of bovine CI with a $Q_{10}$ fully occupying the Q-site (State-α). Residues interacting with $Q_{10}$ in the other three binding states (State-β, -γ and -δ).

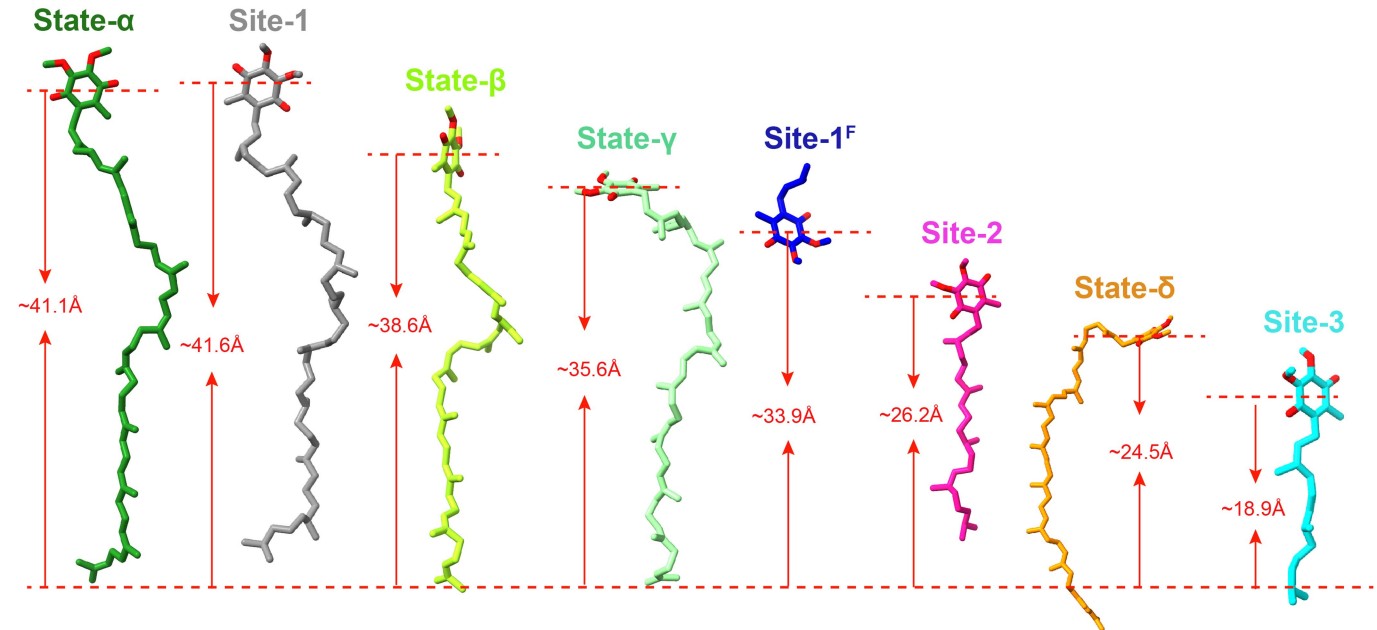

**Extended Data Fig. 8 | Comparison of intermediate binding sites for Q in the Q-channel in this work (α-δ) and previous study (Site1-3).** (PDB: 7VZV, 7W4C, 7W0R, 7W0O). Site-1 is almost identical with State-α, while the other three Q-binding sites (1[F], 2, 3) are not the same as that in ours (state-β, -γ and -δ). The distances shown in the figure are from the head of Q to the entrance of Q-channel.

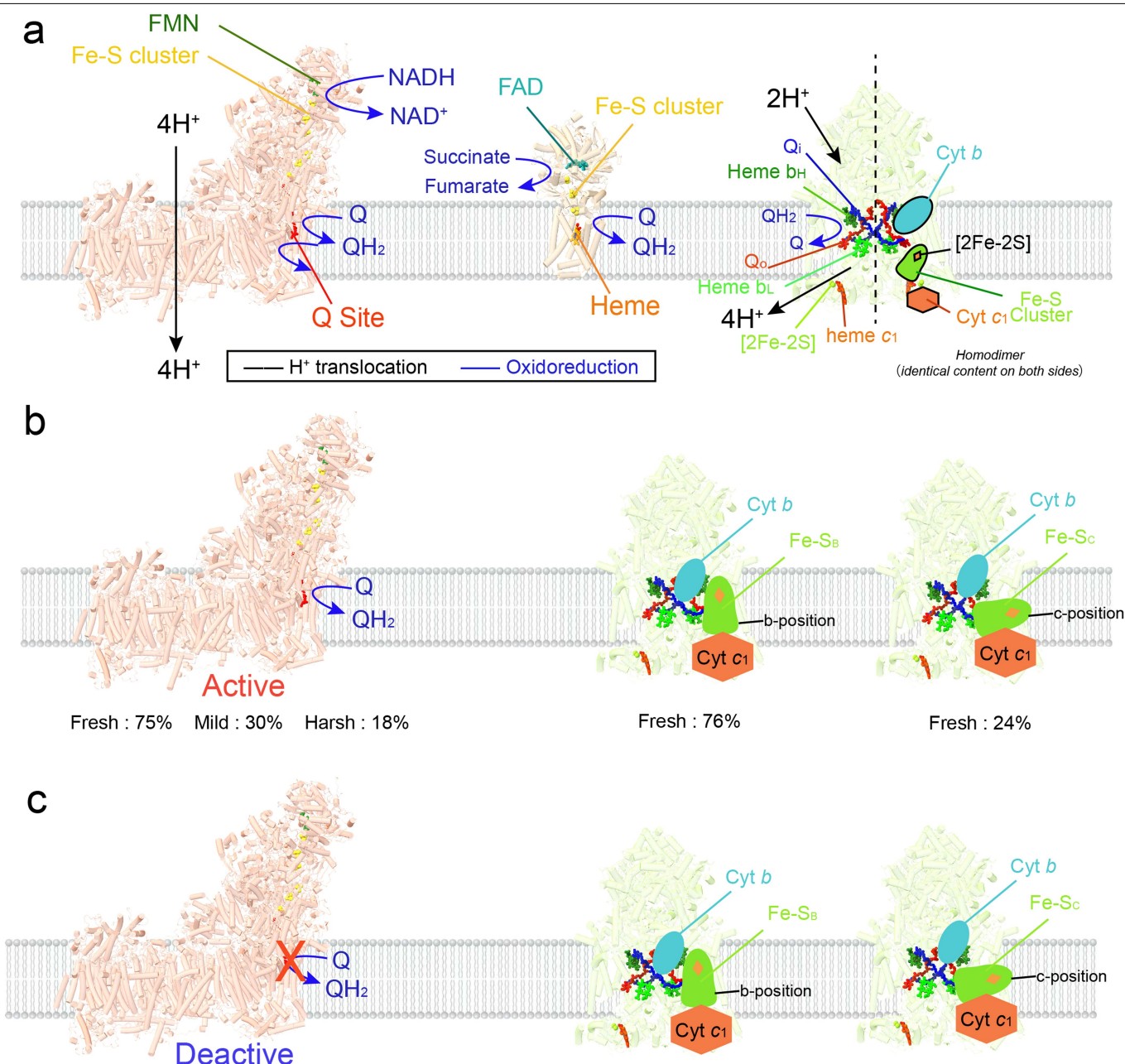

**Extended Data Fig. 9 | Distribution of the conformational states of CI and CIII under different treatments.** a, CIII utilizes $QH_2$ from either CI or II as its substrate. b, In the 'fresh' sample, approximately 75% of CI within SCs is observed to be in an active state. This contrasts with the stability of the active state under varying conditions, where only about 30% and 18% of CI retain their active configuration under 'mild' and 'harsh' experimental conditions, respectively. Notably, in the 'fresh' sample, CI is primarily found in its active form. Simultaneously, within CIII, the ISP domain exhibits a distribution between the b-position (76%) and c-position (24%). c, In the 'fresh' sample, approximately 25% of CI is found in a deactive state. This proportion increases significantly under experimental conditions, with about 70% of CI in a deactive state under 'mild' conditions and further rising to 82% under 'harsh' conditions. Concurrently, the distribution of the ISP domain within CIII in the b-position changes to 60.4% under 'mild' conditions and decreases to 23.3% under 'harsh' conditions. Conversely, the prevalence of the c-position within CIII shifts to 39.6% under 'mild' conditions and escalates to 65.6% under 'harsh' conditions.

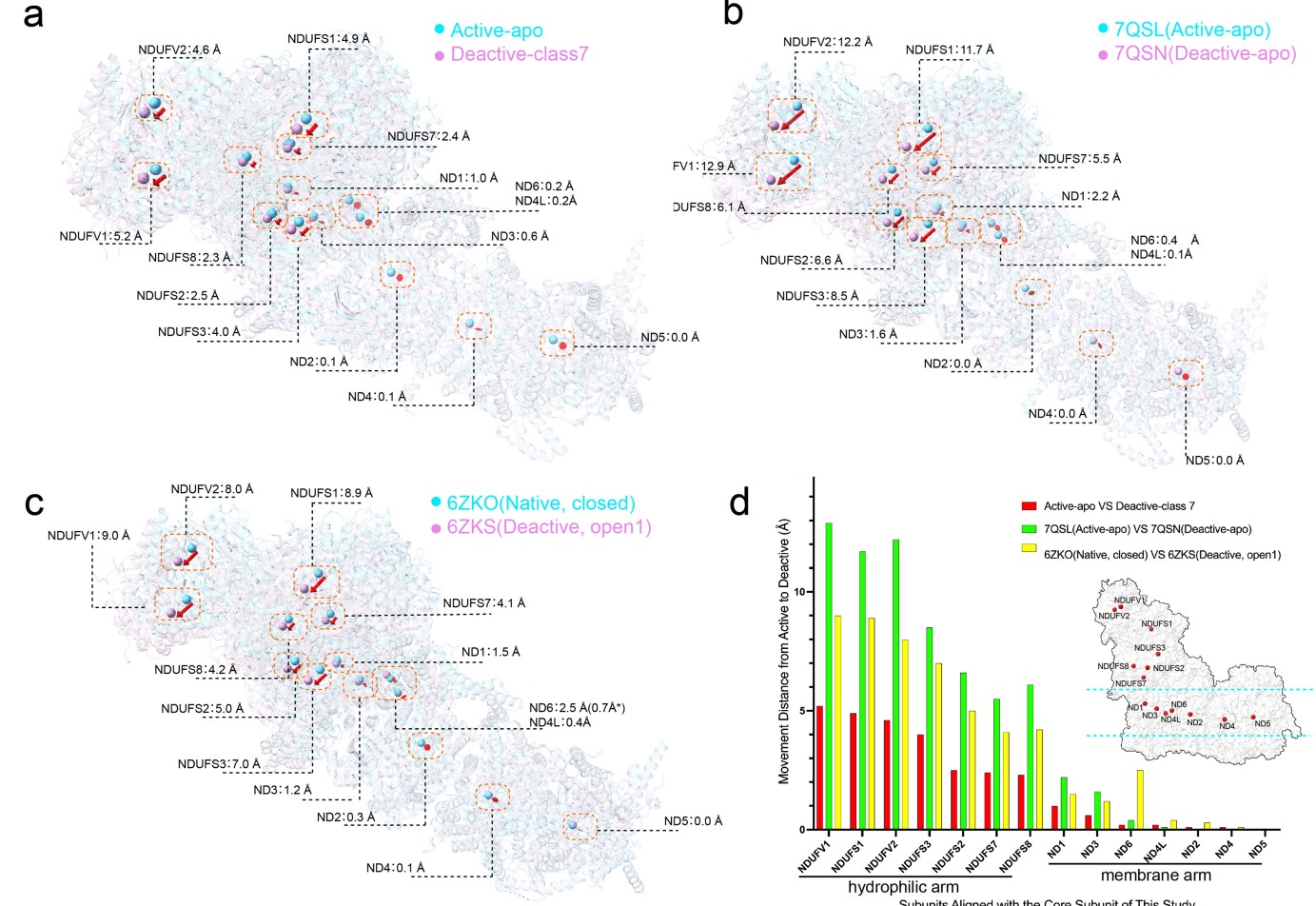

**Extended Data Fig. 10 | Comparison of global conformational changes between active/deactive states from different studies.** For clarity, the centers of the core subunits are used for studying the conformational changes. The centers are calculated by averaging the coordinates of $C_\alpha$ of each subunit. **a-c**, Displacement distances of hydrophilic and hydrophobic core subunits in CI active-apo (cyan) and deactive-class7 (this work, porcine, pink) (**a**); CI active-apo (7QSL, cyan) and deactive-apo (7QSN, bovine, pink) (**b**); and CI Native-closed (6ZKO, cyan) and CI deactive-open1 (6ZKS, ovine, pink) (**c**). **d**, Bar chart statistics of the distances in **a**, **b**, and **c**. *Because the ND6-TMH4 has a big conformational change, this distance comparison is calculated without ND6-TMH4.

# Reporting Summary

## Statistics

For all statistical analyses, confirm that the following items are present in the figure legend, table legend, main text, or Methods section.

| n/a | Confirmed | |
|---|---|---|
| ☐ | ☒ | The exact sample size (*n*) for each experimental group/condition, given as a discrete number and unit of measurement |
| ☐ | ☒ | A statement on whether measurements were taken from distinct samples or whether the same sample was measured repeatedly |
| ☒ | ☐ | The statistical test(s) used AND whether they are one- or two-sided<br>*Only common tests should be described solely by name; describe more complex techniques in the Methods section.* |
| ☒ | ☐ | A description of all covariates tested |
| ☒ | ☐ | A description of any assumptions or corrections, such as tests of normality and adjustment for multiple comparisons |
| ☒ | ☐ | A full description of the statistical parameters including central tendency (e.g. means) or other basic estimates (e.g. regression coefficient) AND variation (e.g. standard deviation) or associated estimates of uncertainty (e.g. confidence intervals) |
| ☒ | ☐ | For null hypothesis testing, the test statistic (e.g. *F*, *t*, *r*) with confidence intervals, effect sizes, degrees of freedom and *P* value noted<br>*Give P values as exact values whenever suitable.* |
| ☒ | ☐ | For Bayesian analysis, information on the choice of priors and Markov chain Monte Carlo settings |
| ☒ | ☐ | For hierarchical and complex designs, identification of the appropriate level for tests and full reporting of outcomes |
| ☒ | ☐ | Estimates of effect sizes (e.g. Cohen's *d*, Pearson's *r*), indicating how they were calculated |

*Our web collection on statistics for biologists contains articles on many of the points above.*

## Software and code

Policy information about availability of computer code

| Data collection | SerialEM 4.1, DigitalMicrograph 3.5. These softwares are also mentioned in the manuscript as well. |
|---|---|
| Data analysis | Relion 4.0, MotionCor2, Gctf 1.08, Gautomatch 0.60, Cryosparc 4.12, Coot 0.9.8.1, Molprobity 4.5.2, PyMOL 2.5.4, ChimeraX 1.5, Phenix 1.20.1, EMAN 2.3, IMOD 4.9.12, https://github.com/builab/subtomo2Chimera, https://github.com/JackZhang-Lab. These softwares/codes are also mentioned in the manuscript as well. |

For manuscripts utilizing custom algorithms or software that are central to the research but not yet described in published literature, software must be made available to editors and reviewers. We strongly encourage code deposition in a community repository (e.g. GitHub). See the Nature Portfolio guidelines for submitting code & software for further information.

## Data

Policy information about availability of data

All manuscripts must include a data availability statement. This statement should provide the following information, where applicable:
- Accession codes, unique identifiers, or web links for publicly available datasets
- A description of any restrictions on data availability
- For clinical datasets or third party data, please ensure that the statement adheres to our policy

All cryo-EM maps and atomic coordinates for mitochondrial supercomplexes have been deposited in the Electron Microscopy Data Bank (EMDB) and the Protein Data Bank(PDB), including entire and locally refined maps of type-A supercomplexes under accession codes EMD-42231/PDB-8UGP, EMD-42225/PDB-8UGH,

EMD-42226/PDB-8UGI; type-B supercomplexes under EMD-42227/PDB-8UGJ; type-O supercomplexes under EMD-42230/PDB-8UGN; type-X supercomplexes under EMD-42233/PDB-8UGR. Different classes of CI under EMD-42165/PDB-8UEO, EMD-42166/PDB-8UEP, EMD-42167/PDB-8UEQ, EMD-42168/PDB-8UER, EMD-42176/PDB-8UEZ, EMD-42169/PDB-8UES, EMD-42170/PDB-8UET, EMD-42171/PDB-8UEU, EMD-42172/PDB-8UEV, EMD-42173/PDB-8UEW, EMD-42174/PDB-8UEX, EMD-42175/PDB-8UEY. Different states of CIII under EMD-42221/PDB-8UGD, EMD-42222/PDB-8UGE, EMD-42223/PDB-8UGF, EMD-42224/PDB-8UGG, EMD-42229/PDB-8UGL. High resolution representative of complex under EMD-42143/PDB-8UD1, EMD-42228/PDB-8UGK, EMD-42229/PDB-8UGL. All these databases/datasets used in the study along with accession-codes are also mentioned in the manuscript under the "Data availability".

## Research involving human participants, their data, or biological material

Policy information about studies with human participants or human data. See also policy information about sex, gender (identity/presentation), and sexual orientation and race, ethnicity and racism.

| | |
|---|---|
| Reporting on sex and gender | N/A |
| Reporting on race, ethnicity, or other socially relevant groupings | N/A |
| Population characteristics | N/A |
| Recruitment | N/A |
| Ethics oversight | N/A |

Note that full information on the approval of the study protocol must also be provided in the manuscript.

# Field-specific reporting

Please select the one below that is the best fit for your research. If you are not sure, read the appropriate sections before making your selection.

☒ Life sciences ☐ Behavioural & social sciences ☐ Ecological, evolutionary & environmental sciences

For a reference copy of the document with all sections, see nature.com/documents/nr-reporting-summary-flat.pdf

# Life sciences study design

All studies must disclose on these points even when the disclosure is negative.

| | |
|---|---|
| Sample size | More than 70,000 cryo-EM movies were collected for final reconstructions. More than one million high-quality particles were included in the final classification and reconstructions of the four different types of supercomplexes. We chose this sample size to achieve the average resolution of type-A supercomplex map better than ~2.5 Å. |
| Data exclusions | Electron microscopy: All micrographs were checked manually, and unusable ones were excluded based on the estimated drift parameters, defocus ranges, estimated usable information by contrast transfer functions, ice thickness, 2D average qualities, 3D classification and reconstruction results using GCTF and cryoSPARC. |
| Replication | Multiple porcine hearts were used in the sample preparations that generated more 50 cryo-EM grids with consistent image qualities to ensure the replication. Cryo-EM structures were lowpass filtered and re-refined, which generated consistent reconstructions. All classified particles were re-processed to generate reference-free 2D class averages that are consistent multiple cycles of classification, demonstrating the replication of our classification approaches. |
| Randomization | All particles were randomly split for the purposes of estimating the resolutions based on the 0.143 criterion of the Fourier Shell Correlation. Otherwise randomization was not relevant to these studies. |
| Blinding | Blinding was not relevant to this study as no populations were preassigned to experimental groups. |

# Reporting for specific materials, systems and methods

We require information from authors about some types of materials, experimental systems and methods used in many studies. Here, indicate whether each material, system or method listed is relevant to your study. If you are not sure if a list item applies to your research, read the appropriate section before selecting a response.

## Materials & experimental systems

| n/a | Involved in the study |
|-----|----------------------|
| ☒ ☐ | Antibodies |
| ☒ ☐ | Eukaryotic cell lines |
| ☒ ☐ | Palaeontology and archaeology |
| ☒ ☐ | Animals and other organisms |
| ☒ ☐ | Clinical data |
| ☒ ☐ | Dual use research of concern |
| ☒ ☐ | Plants |

## Methods

| n/a | Involved in the study |
|-----|----------------------|
| ☒ ☐ | ChIP-seq |
| ☒ ☐ | Flow cytometry |
| ☒ ☐ | MRI-based neuroimaging |

## Plants

| | |
|---|---|
| Seed stocks | *Report on the source of all seed stocks or other plant material used. If applicable, state the seed stock centre and catalogue number. If plant specimens were collected from the field, describe the collection location, date and sampling procedures.* |
| Novel plant genotypes | *Describe the methods by which all novel plant genotypes were produced. This includes those generated by transgenic approaches, gene editing, chemical/radiation-based mutagenesis and hybridization. For transgenic lines, describe the transformation method, the number of independent lines analyzed and the generation upon which experiments were performed. For gene-edited lines, describe the editor used, the endogenous sequence targeted for editing, the targeting guide RNA sequence (if applicable) and how the editor was applied.* |
| Authentication | *Describe any authentication procedures for each seed stock used or novel genotype generated. Describe any experiments used to assess the effect of a mutation and, where applicable, how potential secondary effects (e.g. second site T-DNA insertions, mosiacism, off-target gene editing) were examined.* |

