## [Peer Review File · Nature]

Manuscript Title: High-resolution In-situ Structures of Mammalian Respiratory Supercomplexes

Reviewer Comments & Author Rebuttals

Editorial Notes:

Redactions – unpublished data

Reviewer Reports on the Initial Version:

Referees' comments:

Referee #1 (Remarks to the Author):

This is a very careful and comprehensive study of the structures of various supercomplex formations of respiratory chain complexes from porcine mitochondria using cryo EM single particle analysis and tomography.

There are no dramatic new observations and the novelty here is basically that of comprehensivity with careful analysis of the structures of the different combinations of respiratory chain complexes into supercomplexes, and the way the component complexes interact with one another and with lipids. Some interesting observations are reported on the dynamics of ubiquinone within complexes I and III, but they are also anticipated from earlier work, including molecular dynamics simulations.

My only disappointment in reading this paper concerns the key topic itself - supercomplexes. This is currently a 'hot' issue in the field, but it is not addressed by the present authors, at least not in a very visible way. Yet, it is my opinion that the predated data strongly suggest that 'pairing' of resp. chain complexes into 'super-units' may have little or no other significance than an optimal sharing of limited space in the inner mitochondrial membrane (as important as that is by itself!).

But there seem to be no specific enzyme-functional benefits, such as 'substrate channelling' etc. The individual complexes interact via lipids and thr composition of the assemblies vary widely.

Perhaps the authors might be persuaded to spend a paragrapg of their Discussion on this.

Mårten Wikström

Referee #2 (Remarks to the Author):

The authors of this study succeeded to determine the atomic structures of four different types of respiratory supercomplexes in native membrane environment. This groundbreaking work sets new standards in structural biology.

For the structural analysis of protein complexes in mitochondria, traditional particle selection strategies are insufficient, due to sample thickness, protein crowding in the membrane and strong signal from the membrane. The authors developed a new strategy that is based on the change in membrane structure by the protein inserted into the membrane. Concave local curvature towards the matrix side was found to be associated with the presence of respiratory complex III. Iterative cycles of classifications were started from membrane side views that showed this characteristic distortion of the bilayer. This strategy finally allowed to determine the structure of supercomplexes comprising complexes I, III and IV and associated lipids in native membrane environment at high resolution.

Obviously, this work opens new avenues for studying the structural basis of oxidative phosphorylation in health and disease. The authors address selected aspects in this paper and future work will exploit the huge potential of this new approach. However, the inner mitochondrial membrane is special in that it is packed with a limited number of large enzyme complexes and the question arises which other biological systems could be studied with the methodology developed here.

There is no question that this impressive work should be published in nature. Nevertheless, the authors should consider a few comments:

1. I think it would be appropriate to cite the work of Herrmann Schägger who first described respiratory supercomplexes.
2. The authors have modelled 5,297 water molecules. In the manuscript, protein bound waters are discussed in the context of proton pathways in complex III but not in complex I or complex IV. I suggest to add an SI figure showing water molecules in the transmembrane region of all three complexes.
3. There is an ongoing discussion about the role of supercomplex assembly factors and isoforms of complex IV subunits associated with supercomplex formation. Moreover, there is still some uncertainty about NDUFA4. This protein was initially described as a subunit of complex I but has more recently been assigned to complex IV. Maybe you can comment on this topic.
4. It is described as a noticeable feature that the membrane is bent towards the matrix at the “heel” of complex I. Please note that this has already been reported based on single particle cryoEM and MD simulations (please cite Parey et al. SciAdv 2019 eaax9484).

5. The authors find five major Q binding states in respiratory complex I. This is a major achievement. However, in the way it is presented it appears to be a novel finding that there are intermediate binding sites for Q in the Q tunnel. Maybe you want to cite some more literature here. Along these lines, I find the nomenclature introduced (state 1-5) a bit unfortunate. There is a series of publications that uses the designations site 1 to 5 or similar. It is not clear to me, how e.g. site 4 and state 4 match and there is a clear discrepancy for site 5 and state 5. This may cause confusion.

6. Figure 3 is hard to digest for non-experts. Panel series b will be difficult to read when reduced to final size. I suggest to simplify this figure a bit and make it better readable. Please note a labeling/color code error (ND1/NDUFS7) in panel f state 2. In this context, typo in the text peristatic or peristaltic?

7. Active/deactive transition: Again, it would be fair to cite the publication that described this important feature of complex I for the first time (work of A. Vinogradov). The A/D transition is hotly debated in the complex I field and the results presented here agree well with observations reported by the Hirst group (nicely summarized in Chung et al. Current opinion in structural biology 2022, PMID: 36087446). An alternative view was published by Leo Sazanov (D. Kampjut and L. A. Sazanov, Science 10.1126/science.abc4209 (2020), PMID: 32972993). Do you have indications for or against the relocation of helix 4 of ND6 described by Kampjut and Sazanov? This should be made clear and would help to settle an ongoing dispute on the structural basis of the A/D transition.

8. The authors nicely describe seven hallmarks of A and D form of complex I. Many readers probably first associate the A/D transition with the relative mobility of the two complex I arms and the resulting distinction between “open” and “closed” forms. Do you see the prominent relative movements of both arms of complex I in your native supercomplex structures?

9. The authors claim that they have established conditions of ischemia to characterize the A/D transition of complex I and conformational changes in complex III. The conversion into the D form is known to be promoted by elevated temperature and absence of substrates. In Fig. 5c it is clear that complex I is deactive but what is the status of the Q pool? The figure suggests that in the “fresh” sample the Q pool is largely reduced and it is discussed that under “harsh” conditions it is largely oxidized. This needs some more consideration. I would not assume that the Q pool is simply reduced because complex I is active. In the steady state, electrons flow to oxygen and neither the Q pool nor any of the respiratory complexes will be fully or even highly reduced. In contrast, under conditions of ischemia the Q pool would get reduced because of oxygen limitation. Moreover, at least in principle, it might be possible that the changes observed in complex III are caused by sample treatment independent of the status of complex I and/or the Q pool. Figure 5 and the associated discussion should be carefully reconsidered.

10. Video 2 has a disturbing background noise.

Referee #3 (Remarks to the Author):

In this work Zheng et al developed an in-situ cryo-electron microscopy approach which allows them to determine the high-resolution structures of respiratory supercomplexes (respirasomes) in their native states directly in mitochondrial membrane. They identified four major supercomplex organizations: I1III2IV1, I1III2IV2, I2III2IV2, and I2III4IV2 in the mitochondrial membrane and reported their structures. The supercomplexes I1III2IV2 and I2III4IV2, were not previously identified in the samples purified from mitochondria. So, those structures reported for the first time, are the novel findings. Since the structures are identified in the native mitochondrial membrane the authors found the correlation between the type of the supercomplex and the mitochondrial membrane curvature in the place of SC location. The authors concentrated on the structural details of ubiquinone/ubiquinol exchange mechanism in complex I and the Q-cycle mechanism in complex III as well as the role of phospholipids in organization of these supercomplexes. Importantly, they also for the first time compare the conformations of the respiratory supercomplexes in mitochondria under different environmental redox conditions. The information presented in the manuscript is novel and important for the scientific community.

The authors reported the proportion between 4 types of SCs in the native mitochondrial membrane:

I1III2IV1 (type-A, 64%), I1III2IV2 (type-B, 20%), I2III2IV2, (type-O 0.8%), and I2III4IV2 (type-X, 8%). All together 92.8%. What kind of the structures are represented by the remaining 7.2 %? Does it mean the low % of individual complexes, CIIICIII and CIIICIV SCs in the native membrane? Please explain.

How do Q10 positions and interactions in CI presented in the manuscript correspond to the similar structural data in the previously published papers? Specifically, if compare to the data from the paper of Gu, Liu et al "The coupling mechanism of mammalian mitochondrial complex I". Nat Struct Mol Biol. 2022 Feb;29(2):172-182. The authors of that paper performed cryo-EM study of the purified porcine Complex I and described atomic structures of the respiratory CI in six distinct conditions (Q10, Q10-NADH, decylubiquinone (DQ)-NADH, Q1-NADH, rotenone and rotenone-NADH) at 2.4- to 3.5-Å resolution. The authors of the manuscript do not cite this paper.

From the paper Gu et al:

["Our CI structures of each condition occur in two distinct biochemical states (the classic 'active' or 'deactive' states), although most of the CI particles are in an active state. In the Q chamber and NDUFA9 subunit of the active-state CI structures, we identified three different Q-binding sites along the

Q chamber and a Q-binding pocket in the C-terminal domain of the NDUFA9 subunit. We also detected a Q-binding site in the Q chamber of de-active-state CI”]

This might be included into discussion.

Important note: The discussion part of the manuscript is very brief and rather presents the general conclusions from the result than detailed discussion of the all obtained results.

Fig 5 demonstrates how QH2 produced by CI is used by CIII; in this Fig CI and CIII are not included in the supercomplex. Do authors demonstrate that QH2 produced by CI of one SC is used by CIII in another SC?

Please compare with the model presented in the paper of Letts et al Nature 2016 (your ref. 14).

What about SCs with bound cytochrome c? Are they not present at all?

Not sure that the “comics-like” picture presented in the Fig 3g is appropriate for the publication.

Extended Data Figs. 4-6. Schematic of the workflow for multi-level refinement and focused classification of type-B, type-O, type-X supercomplex. There are only Fig titles without any additional descriptions of the information present in these Figures. Some short description might be helpful.

The content of the paper is interesting for the scientists involved into bioenergetics study; but not all of them are specialists in the cryo-EM and cryo-ET methods. In this case the Fig with workflow for multi-level refinement and focused classification probably should be presented with more clear details, and legends. For example: In the paper of Letts et al and Sazanov. “Structures of Respiratory Supercomplex I+III2 Reveal Functional and Conformational Crosstalk”. Letts et al., 2019, Molecular Cell 75, 1131–1146. SI Fig S4 and S5 “Cryo-EM processing and Focused refinement ” demonstrate this process and provide detailed description of the approach in the Fig legend allowing complete understanding of this process for all readers.

In the Extended Data Figs. 4-6 at many positions one can see: “Globla” / Local CTF”

Please check for typos.

Author Rebuttals to Initial Comments:

Referee #1 (Remarks to the Author):

This is a very careful and comprehensive study of the structures of various supercomplex formations of respiratory chain complexes from porcine mitochondria using cryo EM single particle analysis and tomography.

There are no dramatic new observations and the novelty here is basically that of comprehensivity with careful analysis of the structures of the different combinations of respiratory chain complexes into supercomplexes, and the way the component complexes interact with one another and with lipids. Some interesting observations are reported on the dynamics of ubiquinone within complexes I and III, but they are also anticipated from earlier work, including molecular dynamics simulations.

My only disappointment in reading this paper concerns the key topic itself - supercomplexes. This is currently a 'hot' issue in the field, but it is not addressed by the present authors, at least not in a very visible way. Yet, it is my opinion that the predated data strongly suggest that 'pairing' of resp. chain complexes into 'super-units' may have little or no other significance than an optimal sharing of limited space in the inner mitochondrial membrane (as important as that is by itself!). But there seem to be no specific enzyme-functional benefits, such as 'substrate channelling' etc. The individual complexes interact via lipids and the composition of the assemblies vary widely. Perhaps the authors might be persuaded to spend a paragraph of their Discussion on this.

Mårten Wikström

RESPONSE: We thank the reviewer for the thoughtful consideration of discussion on important open questions such as the enzymatic and functional benefits of the supercomplexes. Even though this is not a simple question that can be straightforwardly addressed using structural approaches alone, we have added a few paragraphs in the discussion section to express our own understanding based on previous studies and current structural results. Here is a summary about our current understanding:

- 1). We agree with this reviewer that it is unlikely that there is a 'direct substrate channeling' between CI and CIII within the same supercomplex as supported by our *in-situ* structures.
- 2). Consistent with previous cryo-ET observations and other evidence, our *in-situ* structures suggest that the localization of these supercomplexes is determined by local membrane curvature, leading to an enrichment of supercomplexes on the planar regions of the cristae. This local enrichment minimizes the distances for substrate and cytochrome *c* diffusion.
- 3). Despite a lack of structural evidence for direct channeling, the special membrane curvature around the local regions of these supercomplexes and their variations may play a role in regulating the Q/QH₂

diffusion and sorting, thus enriching required substrates near Q-sites. Similar concepts have been supported by numerous early studies on lipids and peptides within the membrane.

4). A standard practice in studying supercomplexes functions at the cellular and organism levels *in-vitro* structure guided mutagenesis, blue-native gel and crosslinking mass spectrometry analyses^{1,2}. These studies have provided invaluable insights into the roles and structures of supercomplexes. However, in some of these excellent studies, the authors also clearly acknowledge the limitations of current methods and highlighted the importance of employing more direct structural approaches such as *in-situ* cryo-ET. It is worth noting that we have found significant difference between *in-vitro* structures and *in-situ* supercomplexes in the native membrane environment. Mutagenesis based on *in-vitro* structure need to be compared with future *in-situ* structures in the future. Moreover, conventional approaches might not detect some of the weak associations of these mitochondrial complexes (such as type-X). In the future, a key question is to study the actual forms of mitochondrial complexes *in-situ* in combination with mutagenesis analysis and functional studies.

We have summarized some of our current understanding in the Discussion section.

Referee #2 (Remarks to the Author):

The authors of this study succeeded to determine the atomic structures of four different types of respiratory supercomplexes in native membrane environment. This groundbreaking work sets new standards in structural biology.

For the structural analysis of protein complexes in mitochondria, traditional particle selection strategies are insufficient, due to sample thickness, protein crowding in the membrane and strong signal from the membrane. The authors developed a new strategy that is based on the change in membrane structure by the protein inserted into the membrane. Concave local curvature towards the matrix side was found to be associated with the presence of respiratory complex III. Iterative cycles of classifications were started from membrane side views that showed this characteristic distortion of the bilayer. This strategy finally allowed to determine the structure of supercomplexes comprising complexes I, III and IV and associated lipids in native membrane environment at high resolution.

Obviously, this work opens new avenues for studying the structural basis of oxidative phosphorylation in health and disease. The authors address selected aspects in this paper and future work will exploit the huge potential of this new approach. **However, the inner mitochondrial membrane is special in that it is packed with a limited number of large enzyme complexes and the question arises which other biological systems could be studied with the methodology developed here.**

RESPONSE: We highly appreciate the reviewer's positive comments on the potential impacts and hope our approach can contribute to better understanding of the mitochondrial supercomplex structures and mechanisms within the membrane environment. **[REDACTED]**

We cannot accurately predict the ultimate limit, but we fully believe the new approach will not be limited to respiratory supercomplexes.

There is no question that this impressive work should be published in nature. Nevertheless, the authors should consider a few comments:

1. I think it would be appropriate to cite the work of Herrmann Schagger who first described respiratory supercomplexes.

RESPONSE: We thank the reviewer for the valuable suggestion. We confirm that the reference has been cited. Please refer to citation 28.

2. The authors have modelled 5,297 water molecules. In the manuscript, protein bound waters are discussed in the context of proton pathways in complex III but not in complex I or complex IV. I suggest adding an SI figure showing water molecules in the transmembrane region of all three complexes.

RESPONSE: We have updated our models and made new figures by following the reviewer's suggestion. Please refer to Extended Data Fig. 9.

Extended Data Fig. 9 | Water molecules in CI, CIII, and CIV.

a-c, The water molecules bound in CI, CIII and CIV. Blue dots represent water molecules in the hydrophilic regions, while red dots indicate water molecules near the hydrophobic regions. The core subunits of CI, CIII and CIV are color-coded as shown in the figure.

3. There is an ongoing discussion about the role of supercomplex assembly factors and isoforms of complex IV subunits associated with supercomplex formation. Moreover, there is still some uncertainty about NDUFA4. This protein was initially described as a subunit of complex I but has more recently been assigned to complex IV. Maybe you can comment on this topic.

RESPONSE: We thank the reviewer for this very insightful question. In our current study, the structures determined from the porcine heart mitochondria are dominantly in the mature forms, and therefore we have not identified the assembly factors which co-exist in the supercomplexes during the assembling process. This is potentially an exciting question to be explored using different sources of mitochondria in the future. Furthermore, we have only identified a specific set of the isoforms of CIV subunits in our current structures. It is still possible that other isoforms of the CIV subunits co-exist. However, the expression levels of different isoforms have been reported to be tissue dependent³. Therefore, the

abundance of other isoforms is expected to be significantly less than the subunits we currently built using the porcine heart mitochondria, and our classification has not yet allowed us to separate the rare isoforms from the dominant ones. In addition, the resolutions of CIV' in type-B and the two CIV in type-X are not sufficient to distinguish them from the dominant CIV isoforms in type-A. In the future work, this will be a highly valuable question to address using mitochondria from diverse tissue samples to study how different isoforms distribute and in turn affect supercomplex formation, and how they might be linked to the regulation of supercomplex functions. We have provided a table to address this question. Please refer to Supplementary Table 1 and page 5, line 133-141.

Chain ID	Gene Name	isoform
4A	coxI	
4B	coxII	
4C	coxIII	
4D*	cox4	COX4 isoform 1
4E	cox5A	
4F	cox5B	
4G*	cox6A	COX6A2
4H*	cox6B	COX 6B1 isoform X1
4I	cox6C	
4J*	cox7A	COX 7A1
4K*	cox7B	COX 7B
4L	cox7C	
4M*	cox8	COX8H
4N*	ndufa4	NDUFA4

Supplementary Table 1 | Isoform identification for CIV subunits.

For subunits with isoforms, we identify in the structure using sidechain features and list the specific isoforms in the table.

*These subunits have isoforms.

Regarding the NDUFA4 subunit, we confirm that in all the four types of *in-situ* supercomplex structures, it is localized in CIV and far from CI. Our conclusion is consistent with a recent study by Zong et al⁴. Moreover, this subunit blocks the CIV dimerization interfaces⁴ as revealed in early crystallographic studies. We have not observed similar CIV dimers in our *in-situ* structures. We speculate that the NDUFA4 subunit can fall off CIV to allow the dimerization of CIV under certain conditions such as environmental stress. Please refer to Extended Data Fig. 15 and page 5, line 133-141.

Extended Data Fig. 15 | Localization of NDUFA4 in the *in-situ* respiratory supercomplex. a, b, c,d, Localization of NDUFA4 (red) in type-A, B, O, and X supercomplexes. Complex I is shown in pale red, Complex III₂ in yellow, and Complex IV in blue. NDUFA4 is positioned on CIV, distant from CI in all supercomplex types.

4. It is described as a noticeable feature that the membrane is bent towards the matrix at the “heel” of complex I. Please note that this has already been reported based on single-particle cryoEM and MD simulations (please cite Parey et al. SciAdv 2019 eaax9484).

RESPONSE: We thank the reviewer and have included the citation in the revised manuscript. Please refer to citation 19.

5. The authors find five major Q binding states in respiratory complex I. This is a major achievement. However, in the way it is presented it appears to be a novel finding that there are intermediate binding sites for Q in the Q tunnel. Maybe you want to cite some more literature

here. Along these lines, I find the nomenclature introduced (state 1-5) a bit unfortunate. There is a series of publications that uses the designations site 1 to 5 or similar. It is not clear to me, how e.g. site 4 and state 4 match and there is a clear discrepancy for site 5 and state 5. This may cause confusion.

RESPONSE: We appreciate the reviewer's valuable suggestions on the careful nomenclature of different states and citation of previous publications. We used different nomenclature because some of the five states (apo and α , β , γ , δ) are not exactly the same as described in the previously reported *in-vitro* structures by Gu et al⁵. Specifically, the apo state and Q-binding site 1 are consistent with our structures in the state-apo and $-\alpha$ respectively. The other three Q-binding sites (1^F, 2, 3) are not the same as that in ours (state- β , $-\gamma$ and $-\delta$). To avoid possible confusion, we decided to use different nomenclature to reflect their relative positions in a possibly sequential manner. Since this is a highly dynamic process, we suppose the different Q-binding sites from both Gu et al⁵ and ours may present different intermediate states⁶. Future structural work is still required to reveal a more comprehensive landscape of intermediate Q-binding states to fully understand the dynamic process of Q/QH₂ turnover within CI. Please refer to Extended Data Fig. 16 and page 5, line 154-156.

Extended Data Fig. 16 | Comparison of intermediate binding sites for Q in the Q-channel in this work ($-\delta$) and previous study (Site1-3) (PDB: 7VZV, 7W4C, 7W0R, 7W00). Site-1 is almost identical with State- α , while the other three Q-binding sites (1^F, 2, 3) are not the same as that in ours (state- β , $-\gamma$ and $-\delta$). The distances shown in the figure are from the head of Q to the entrance of Q-channel.

6. Figure 3 is hard to digest for non-experts. Panel series b will be difficult to read when reduced to final size. I suggest to simplify this figure a bit and make it better readable. Please note a labeling/color code error (ND1/NDUFS7) in panel f state 2. In this context, typo in the text peristatic or peristaltic?

RESPONSE: We have revised Fig. 3 by moving panel e and f to Extended Data Fig. 17. We double checked the color code in panel f state 2. Because the Q-site was rotated for better presentation, the labels in this panel might have led to the confusion. We confirm the color codes in all panels now align with each other. We also thank the reviewer for pointing out the typo. It has been corrected.

Fig. 3 | High-resolution *in-situ* structures reveal multiple Q/QH₂ binding states within the Q-binding channel.

a, Cartoon models of CI active-apo state, with subunits constituting the Q-binding channel highlighted. **b**, Variations in distances between the Q₁₀ headgroup and membrane surfaces across fully occupied, two intermediate, and half-occupied states. **c**, Spatial positioning of Q₁₀ within the Q-binding channel for the four distinct binding states. **d**, Comparisons of three other binding states

with the fully occupied Q₁₀ (transparent stick). The distance from the Q₁₀ headgroup to the quinone-binding channel entrance varies among different binding states. **e**, Schematic depiction of the silkworm-like undulatory motion of Q₁₀ within the Q-binding channel.

Extended Data Fig. 17 | Different Q-binding states and conformational changes in the Q-channel.

a, Superpositions of the four different Q-binding states with the active-apo state (grey) reveal not only conformational alterations in the Q₁₀ headgroup and H59^{NDUFS2} (left panel) but also significant changes in long-range structures away from Q₁₀ headgroup (right panel). **b-e**, Superimposition of the porcine *in-situ* supercomplex structure with the bovine CI (grey) incorporated into MSP nanodisks indicates that our fully occupied state resembles the active form of bovine CI with a Q₁₀ fully occupying the Q-site (State- α). Residues interacting with Q₁₀ in the other three binding states (State- β , - γ and - δ) .

7. Active/deactive transition: Again, it would be fair to cite the publication that described this important feature of complex I for the first time (work of A. Vinogradov). The A/D transition is hotly debated in the complex I field and the results presented here agree well with observations reported by the Hirst group (nicely summarized in Chung et al. Current opinion in structural biology 2022, PMID: 36087446). An alternative view was published by Leo Sazanov (D. Kampjut and L. A. Sazanov, Science 10.1126/science.abc4209 (2020), PMID: 32972993). Do you have indications for or against the relocation of helix 4 of ND6 described by Kampjut and Sazanov? This should be made clear and would help to settle an ongoing dispute on the structural basis of the A/D transition.

RESPONSE: We highly appreciate the reviewer's insightful comments. We have added several relevant citations, including A. [REDACTED] Our extensive 3D classification of the *in-situ* supercomplex structures has not yet revealed a state that resembles the relocation of the helix 4 of ND6. In particular, our structures in the native membrane environment further imply that the relocation mechanism proposed by Kampjut et al is unlikely to occur. Because the conformational change of this transmembrane helix is so huge, it will have to cause collective relocations of many surrounding lipid molecules, which will substantially change the local protein-lipids interactions. Considering that the goal of the whole process is to reduce Q to QH₂ by transferring two electrons and that the reaction is extremely efficient and fast, the relocation mechanism that involves a substantial reorganization of the local membrane environment seems too costly. In summary, our *in-situ* results are in favor of the model proposed by the Hirst group in this case. Please refer to Extended Data Fig. 22 and Discussion Session.

Extended Data Fig. 22 | Comparison conformational changes of CI ND6-TMH4 during active/deactive transition across different species.

a-c, Comparison of CI ND6-TMH4 between active and deactive states from different species. The active states are shown in gray: active-apo (this work, porcine), active-apo (PDB ID: 7QSL, bovine), and native-closed (PDB ID: 6ZKO, ovine). The deactive states are shown in color: deactive-class7 (this work, porcine, yellow), deactive-apo (PDB ID: 7QSN, bovine, blue), and deactive-open1 (PDB ID: 6ZKS, ovine, orange). The relocation of TMH4 is only observed in ovine.

d-f, Comparison of CI ND6-TMH4 deactive states between porcine and bovine (**d**), porcine and ovine (**e**), and bovine and ovine (**f**).

8. The authors nicely describe seven hallmarks of A and D form of complex I. Many readers probably first associate the A/D transition with the relative mobility of the two complex I arms and the resulting distinction between “open” and “closed” forms. Do you see the prominent relative movements of both arms of complex I in your native supercomplex structures?

RESPONSE: Yes, we confirm that the relative movements of both arms of CI A/D states can be clearly observed. We have also quantified the movement and provided a supporting figure, Extended Data Fig. 21, to demonstrate the conformational changes between A/D states. Notably, the relative movement between the two arms seem to be much smaller in the native environment than those reported structures in the MSP-nanodisc⁶ and in the detergent micelle⁷, please refer to Discussion Session.

Extended Data Fig. 21 | Comparison of global conformational changes between active/deactive states from different studies.

For clarity, the centers of the core subunits are used for studying the conformational changes. The centers are calculated by averaging the coordinates of C_α of each subunit. **a-c**, Displacement

distances of hydrophilic and hydrophobic core subunits in CI active-apo (cyan) and deactive-class7 (this work, porcine, pink) **(a)**, CI active-apo (7QSL, cyan) and deactive-apo (7QSN, pink) (bovine) **(b)**, and CI Native-closed (6ZKO, cyan) and CI deactive-open1 (6ZKS) (ovine, pink) **(c)**. **d**, Bar chart statistics of the distances in **a**, **b**, and **c**.

*Because the ND6-TMH4 has a big conformational change, this distance comparison is calculated without ND6-TMH4.

9. The authors claim that they have established conditions of ischemia to characterize the A/D transition of complex I and conformational changes in complex III. The conversion into the D form is known to be promoted by elevated temperature and absence of substrates. In Fig. 5c it is clear that complex I is deactive but what is the status of the Q pool? The figure suggests that in the “fresh” sample the Q pool is largely reduced and it is discussed that under “harsh” conditions it is largely oxidized. This needs some more consideration. I would not assume that the Q pool is simply reduced because complex I is active. In the steady state, electrons flow to oxygen and neither the Q pool nor any of the respiratory complexes will be fully or even highly reduced. In contrast, under conditions of ischemia the Q pool would get reduced because of oxygen limitation. Moreover, at least in principle, it might be possible that the changes observed in complex III are caused by sample treatment independent of the status of complex I and/or the Q pool. Figure 5 and the associated discussion should be carefully reconsidered.

RESPONSE: First of all, we appreciate the insightful question posed by the reviewer. We acknowledge that during ischemic conditions, the oxygen supply is limited, potentially leading to a more reduced Q pool. Our original intention was to highlight the hypothesis that the 'local QH₂ concentration' produced by complex I is diminished, rather than suggesting changes in the overall Q pool oxidation status. In our findings, the conformational distributions of complexes I and III are well-defined; however, the possibility of direct conformational coupling between these complexes remains uncertain. We agree with the reviewer that the conformation of complex III may be directly influenced by varying treatments, and we have accordingly adjusted our proposed model. To prevent over-interpretation of our current observations, we have revised the hypothesis regarding the coupled conformational changes between I and III in a functional perspective, and simply stated that there is an observed correlation under specific conditions, please refer to page 9, line 262-278.

10. Video 2 has a disturbing background noise.

RESPONSE: We confirm that the background noise of Video 2 has been removed.

Referee #3 (Remarks to the Author):

In this work Zheng et al developed an in-situ cryo-electron microscopy approach which allows them to determine the high-resolution structures of respiratory supercomplexes (respirasomes) in their native states directly in mitochondrial membrane. They identified four major supercomplex organizations: $I_1III_2IV_2$, $I_2III_4IV_2$, $I_1III_2IV_1$, and $I_2III_4IV_1$ in the mitochondrial membrane and reported their structures. The supercomplexes $I_1III_2IV_1$ and $I_2III_4IV_1$ were not previously identified in the samples purified from mitochondria. So, those structures reported for the first time, are the novel findings. Since the structures are identified in the native mitochondrial membrane the authors found the correlation between the type of the supercomplex and the mitochondrial membrane curvature in the place of SC location. The authors concentrated on the structural details of ubiquinone/ubiquinol exchange mechanism in complex I and the Q-cycle mechanism in complex III as well as the role of phospholipids in organization of these supercomplexes. Importantly, they also for the first time compare the conformations of the respiratory supercomplexes in mitochondria under different environmental redox conditions. The information presented in the manuscript is novel and important for the scientific community.

1. The authors reported the proportion between 4 types of SCs in the native mitochondrial membrane:

$I_1III_2IV_1$ (type-A, 64%), $I_1III_2IV_2$ (type-B, 20%), $I_2III_4IV_2$, (type-O 0.8%), and $I_2III_4IV_1$ (type-X, 8%). All together 92.8%. What kind of the structures are represented by the remaining 7.2 %? Does it mean the low % of individual complexes, CIIICIII and CIIICIV SCs in the native membrane? Please explain.

RESPONSE: There are indeed lower frequencies of other forms of supercomplexes as indicated by weak densities, such as type-O dimer ($I_4III_4IV_4$). Since it is not feasible to estimate the actual number of those unresolved states, we only quantified what could be clearly visualized in this case. We are sorry for the confusion. It was indeed “type-O,8%” in Fig. 2, and we missed a space between “,” and “8”, so it looked like 0.8%. We have put a space there and it is now “type-O, 8%”.

2. How do Q10 positions and interactions in CI presented in the manuscript correspond to the similar structural data in the previously published papers? Specifically, if compare to the data from the paper of Gu, Liu et al “The coupling mechanism of mammalian mitochondrial complex I”. Nat Struct Mol Biol. 2022 Feb;29(2):172-182. The authors of that paper performed cryo-EM study of the purified porcine Complex I and described atomic structures of the respiratory CI in six distinct conditions (Q10, Q10–NADH, decylubiquinone (DQ)–NADH, Q1–NADH, rotenone and rotenone–NADH) at 2.4- to 3.5-Å resolution. The authors of the manuscript do not cite this paper. From the paper Gu et al: [“Our CI structures of each condition occur in two distinct biochemical states (the classic ‘active’ or ‘deactive’ states), although most of the CI particles are in an active state. In the Q chamber and NDUFA9 subunit of the active-state CI structures, we identified three different Q-binding sites along the Q chamber and a Q-binding pocket in the C-terminal domain of the NDUFA9 subunit. We also detected a Q-binding site in the Q chamber of de-active-state CI”] This might be included into discussion.

Important note: The discussion part of the manuscript is very brief and rather presents the general conclusions from the result than detailed discussion of all obtained results.

RESPONSE: First, we highly appreciate the reviewer's valuable suggestion and have compared our structures with those reported by Gu et al. Similar questions have been raised by another reviewer. Specifically, in both papers, there are two states that are essentially the same: the apo state and the Q-site 1. Other states cannot be directly compared. This is not a surprise because there are potentially many intermediate states. Therefore, we speculate that different intermediate states have been captured in the two studies, except the apo and Q-site 1. We confirmed that the published papers on the different Q-binding sites have been cited, compared, and discussed in the revised manuscript. Please refer to Extended Data Fig. 16 and page 5, line 154-156.

Extended Data Fig. 16 | Comparison of intermediate binding sites for Q in the Q-channel in this work (- δ) and previous study (Site1-3) (PDB: 7VZV, 7W4C, 7W0R, 7W00). Site-1 is almost identical with State- α , while the other three Q-binding sites (1^F, 2, 3) are not the same as that in ours (state- β , - γ and - δ). The distances shown in the figure are from the head of Q to the entrance of Q-channel.

In our work, all different Q/QH₂-binding states are mostly in the closed state except state- β shows open conformation with the only hallmark TMH1-2 of ND3 disordered. Please refer to Extended Data Fig. 18.

Extended Data Fig. 18 | Analysis of the hallmarks of the four distinct Q-occupied states.

State- α , - γ , - δ exhibit standard active features, while State- β show partial deactive features, suggesting these conformational changes may be required for the intermediate states during the reaction.

3. Fig 5 demonstrates how QH2 produced by CI is used by CIII; in this Fig CI and CIII are not included in the supercomplex. Do authors demonstrate that QH2 produced by CI of one SC is used by CIII in another SC?

Please compare with the model presented in the paper of Letts et al Nature 2016 (your ref. 14).

RESPONSE: We revised the discussion about the Fig. 5, however from our study we think QH2 produced by CI is not necessarily used by CIII in the supercomplex.

4. What about SCs with bound cytochrome c? Are they not present at all?

RESPONSE: This is really an excellent question. We have tried to classify all possible states that might display clear cytochrome c density, but unfortunately, have not obtained positive results so far. We suspect SC-cyt c binding is highly transient within the mitochondrial intermembrane space, and only occurs during the rapid electron transfer process.

5. Not sure that the “comics-like” picture presented in the Fig 3g is appropriate for the publication.

RESPONSE: Maybe we can leave it to the editor to decide whether it is appropriate.

6. Extended Data Figs. 4-6. Schematic of the workflow for multi-level refinement and focused classification of type-B, type-O, type-X supercomplex. There are only Fig titles without any additional descriptions of the information present in these Figures. Some short description might be helpful. The content of the paper is interesting for the scientists involved into bioenergetics study; but not all of them are specialists in the cryo-EM and cryo-ET methods. In this case the Fig with workflow for multi-level refinement and focused classification probably should be presented with more clear details, and legends. For example: In the paper of Letts et al and Sazanov. “Structures of Respiratory Supercomplex I+III₂ Reveal Functional and Conformational Crosstalk”. Letts et al., 2019, *Molecular Cell* 75, 1131–1146. SI Fig S4 and S5 "Cryo-EM processing and Focused refinement " demonstrate this process and provide detailed description of the approach in the Fig legend allowing complete understanding of this process for all readers.

RESPONSE: We have added more detailed legends describing the process, Please refer to Extended Data Fig. 3-6.

7. In the Extended Data Figs. 4-6 at many positions one can see: “Globla” / Local CTF” Please check for typos.

RESPONSE: We thank the reviewer for pointing out the typos. We have corrected them.

- 1 Milenkovic, D. *et al.* Preserved respiratory chain capacity and physiology in mice with profoundly reduced levels of mitochondrial respirasomes. *Cell Metab* **35**, 1799-1813 e1797, doi:10.1016/j.cmet.2023.07.015 (2023).
- 2 Lapuente-Brun, E. *et al.* Supercomplex assembly determines electron flux in the mitochondrial electron transport chain. *Science* **340**, 1567-1570, doi:10.1126/science.1230381 (2013).
- 3 Sinkler, C. A. *et al.* Tissue- and Condition-Specific Isoforms of Mammalian Cytochrome c Oxidase Subunits: From Function to Human Disease. *Oxid Med Cell Longev* **2017**, 1534056, doi:10.1155/2017/1534056 (2017).

- 4 Zong, S. *et al.* Structure of the intact 14-subunit human cytochrome c oxidase. *Cell Res* **28**, 1026-1034, doi:10.1038/s41422-018-0071-1 (2018).
- 5 Gu, J., Liu, T., Guo, R., Zhang, L. & Yang, M. The coupling mechanism of mammalian mitochondrial complex I. *Nat Struct Mol Biol* **29**, 172-182, doi:10.1038/s41594-022-00722-w (2022).
- 6 Chung, I. *et al.* Cryo-EM structures define ubiquinone-10 binding to mitochondrial complex I and conformational transitions accompanying Q-site occupancy. *Nat Commun* **13**, 2758, doi:10.1038/s41467-022-30506-1 (2022).
- 7 Kampjut, D. & Sazanov, L. A. The coupling mechanism of mammalian respiratory complex I. *Science* **370**, doi:10.1126/science.abc4209 (2020).

Reviewer Reports on the First Revision:

Referees' comments:

Referee #2 (Remarks to the Author):

The authors addressed all my questions and the revised manuscript is now ready for publication. A few typos need correction and I suggest to consider a few points that could still be improved:

Line 156: Literature on Q binding sites in complex I is still incomplete, e.g. you don't mention any of the MD simulation papers on this topic.

Line 196: Please state in the text in which sample type the seven classes were identified. This should also be explained in the legend of supplementary table 2. I suppose it is the "fresh" sample but please indicate this clearly.

Line 286: Please add ref 7 (Schägger and Pfeiffer, EMBO J 2000). H.S. invented BN PAGE and described supercomplexes for the first time. Please also check correct spelling of the Journal in the reference list.

Line 345: "loss of a normal physiological environment" could be misunderstood as denaturation. The A/D transition is reversible.

Line 346: The last sentence of the paragraph is difficult to read and needs some editing (see also typo close/closed)

Reference list, ref 24 erase "Structural Biology"

Fig. 3b: The level of the mitochondrial membrane is the "average" level. Lipid headgroups shown are obviously not consistent because of local membrane distortion.

Fig 5: The connection between different states of complex I and complex III as shown in Fig. 5 is still a bit vague and inconclusive. Does the information presented really need a figure in the main text? It might be possible to transfer this figure to SI.

Extended data Fig. 17: Check panels C and D. I am still quite confident that you have mixed up NDFUS7 and ND1.

Referee #3 (Remarks to the Author):

I looked through the revised manuscript and authors response to reviewers' comments. My opinion is that the points raised in the previous round of review were satisfactorily addressed by the authors, and the manuscript should be published.

Author Rebuttals to First Revision:

Line 156: Literature on Q binding sites in complex I is still incomplete, e.g. you don't mention any of the MD simulation papers on this topic.

RESPONSE: We have added this citation to the corresponding section and updated the text.

Line 196: Please state in the text in which sample type the seven classes were identified. This should also be explained in the legend of supplementary table 2. I suppose it is the "fresh" sample but please indicate this clearly.

RESPONSE: The classification analysis was performed using combined datasets. We have clarified in the supplementary table.

Line 286: Please add ref 7 (Schägger and Pfeiffer, EMBO J 2000). H.S. invented BN PAGE and described supercomplexes for the first time. Please also check correct spelling of the Journal in the reference list.

RESPONSE: We have added the reference. We also have checked the reference list.

Line 345: "loss of a normal physiological environment" could be misunderstood as denaturation. The A/D transition is reversible.

RESPONSE: We appreciate this clarification and have revised the expression in this section.

Line 346: The last sentence of the paragraph is difficult to read and needs some editing (see also typo close/closed)

RESPONSE: We thank the reviewer for pointing out the typo and confirm it has been corrected.

Reference list, ref 24 erase "Structural Biology"

RESPONSE: We really appreciate the careful reading of this reviewer, and confirm that we have checked and updated the reference list. Please refer to citation 27.

Fig. 3b: The level of the mitochondrial membrane is the "average" level. Lipid headgroups shown are obviously not consistent because of local membrane distortion.

RESPONSE: We thank the reviewer for pointing out this issue. The lower dashed line actually indicated the middle of the lipid bilayer in our last version of Fig 3b. We agree this was confusing. The membrane layer here specifically presents the local region near the Q-binding pocket, i.e. the membrane surrounding the complex I 'heel'. We have clarified this in the revised legends and also moved down the lower line to make sure this aligns with the lower surface of the membrane. Please refer to Fig. 3b legends.

Fig 5: The connection between different states of complex I and complex III as shown in Fig. 5 is still a bit vague and inconclusive. Does the information presented really need a figure in the main text? It might be possible to transfer this figure to SI.

RESPONSE: We feel this is still an open question and needs further investigation. We also agree with the reviewer's opinion that this could be simply led by the treatment itself. Therefore, we have decided to tone down our original statement that the conformational states of CI and CIII are 'coupled'. We have already moved Fig 5 to the Extended Data 9.

Extended data Fig. 17: Check panels C and D. I am still quite confident that you have mixed up NDFUS7 and ND1.

RESPONSE: We thank this reviewer and confirm the reviewer is correct. The labels have been updated.

- 1 Milenkovic, D. *et al.* Preserved respiratory chain capacity and physiology in mice with profoundly reduced levels of mitochondrial respirasomes. *Cell Metab* **35**, 1799-1813 e1797, doi:10.1016/j.cmet.2023.07.015 (2023).
- 2 Lapuente-Brun, E. *et al.* Supercomplex assembly determines electron flux in the mitochondrial electron transport chain. *Science* **340**, 1567-1570, doi:10.1126/science.1230381 (2013).
- 3 Sinkler, C. A. *et al.* Tissue- and Condition-Specific Isoforms of Mammalian Cytochrome c Oxidase Subunits: From Function to Human Disease. *Oxid Med Cell Longev* **2017**, 1534056, doi:10.1155/2017/1534056 (2017).
- 4 Zong, S. *et al.* Structure of the intact 14-subunit human cytochrome c oxidase. *Cell Res* **28**, 1026-1034, doi:10.1038/s41422-018-0071-1 (2018).
- 5 Gu, J., Liu, T., Guo, R., Zhang, L. & Yang, M. The coupling mechanism of mammalian mitochondrial complex I. *Nat Struct Mol Biol* **29**, 172-182, doi:10.1038/s41594-022-00722-w (2022).
- 6 Chung, I. *et al.* Cryo-EM structures define ubiquinone-10 binding to mitochondrial complex I and conformational transitions accompanying Q-site occupancy. *Nat Commun* **13**, 2758, doi:10.1038/s41467-022-30506-1 (2022).
- 7 Kampjut, D. & Sazanov, L. A. The coupling mechanism of mammalian respiratory complex I. *Science* **370**, doi:10.1126/science.abc4209 (2020).